# Aberrant choroid plexus formation drives the development of treatment-related brain toxicity
Tamara Bender [1,7], Esther Schickel [1,7], Celine Schielke[1], Jürgen Debus[2,3], David R. Grosshans [4 ✉], Marco Durante [1,5,6 ✉] & Insa S. Schroeder [1 ✉]

Brain tumors are commonly treated with radiotherapy, but the efficacy of the treatment is limited by its toxicity to the normal tissue including post-irradiation contrast enhanced lesions often linked to necrosis. The poorly understood mechanisms behind such brain lesions were studied using cerebral organoids. Here we show that irradiation of such organoids leads to dose-dependent growth retardation and formation of liquid-filled cavities but is not correlated with necrosis. Instead, the radiation-induced changes comprise of an enhancement of cortical hem markers, altered neuroepithelial stem cell differentiation, and an increase of ZO1$^+$/AQP1$^+$/CLDN3$^+$-choroid plexus (CP)-like structures accompanied by an upregulation of *IGF2* mRNA, known to be expressed in CP and cerebrospinal fluid. The altered differentiation is attributed to changes in the WNT/BMP signaling pathways. We conclude that aberrant CP formation can be involved in radiation-induced brain lesions providing additional strategies for possible countermeasures.

Primary brain tumors, which may arise in pediatric or adult patients are commonly treated with cranial irradiation. Historical studies have suggested improved disease control with dose-escalated radiation therapy employing either photons (X-rays) or charged particles (e.g. protons). However, dose escalation is hampered by the development of radiation necrosis (RN)[1–4], a common side effect in many types of brain tumors including low-grade and high-grade glioma, meningioma, and metastatic brain tumors[5–9]. In pediatric patients, low-grade gliomas including pilocytic astrocytoma and ependymoma, or other tumor types such as medulloblastoma are more common than high-grade glioma, and patients are often treated with high-energy protons in order to spare normal brain tissues to a greater extent than X-rays do[10]. However, clinical studies raised concerns that pediatric brain tumor patients treated with protons experienced unexpectedly high rates of RN[11,12], and a variety of other radiation-induced lesions[13].

RN varies in severity and is often associated with contrast-enhancing lesions (CEL) in post-treatment magnetic resonance imaging (MRI) scans. RN is diagnosed in up to 25% of the patients treated with cranial radiation[14]. For symptomatic patients, bevacizumab has been shown to lower the rate of RN[15] and is recommended in several guidelines. However, the diagnosis and treatment of RN remains extremely challenging as the observed CEL in brain tumor patients may represent a variety of pathophysiologies including RN, blood-brain barrier disruption (BBD), or tumor progression. Necrosis and progression occur commonly in recurrent glioblastoma, and brain metastases[16], and often cannot be reproducibly diagnosed by pathologists[17]. Both BBD and RN are believed to stem from radiation-induced vascular damage leading to local ischemia and hypoxia that will eventually induce necrosis or angiogenesis and subsequent brain edema due to increased permeability of the newly formed blood vessels. While BBD occurs within the first six months of radiotherapy, RN usually occurs 6-18 months after radiation exposure but has also been observed in pediatric patients as early as 1.2 months (median time to onset[18]). Within the context of stereotactic radiotherapy, this sequela can present years or decades after radiotherapy[19]. BBD is generally believed to be transient but can transition into RN. Progressive, untreated RN, which is characterized by large edematous lesions and pronounced clinical symptoms, maybe both irreversible and lethal[19].

Contemporary clinical studies do not offer clear insights into the onset and progression of RN and to the classification of CEL as necrosis. We elected to use cerebral organoids from human pluripotent stem cells as a

[1]GSI Helmholtzzentrum für Schwerionenforschung, Biophysics Department, Darmstadt, Germany. [2]Heidelberg University, Faculty of Medicine, and Heidelberg Ion-Beam Therapy Center (HIT), Heidelberg, Germany. [3]Clinical Cooperation Unit Radiation Oncology, German Cancer Research Center (DKFZ), Heidelberg, Germany. [4]Department of Radiation Oncology, Division of Radiation Oncology, The University of Texas MD Anderson Cancer Center, Houston, TX, USA. [5]Institute for Condensed Matter Physics, Technische Universität Darmstadt, Darmstadt, Germany. [6]Department of Physics „Ettore Pancini", University Federico II, Naples, Italy. [7]These authors contributed equally: Tamara Bender, Esther Schickel. ✉e-mail: dgrossha@mdanderson.org; marco.durante@tu-darmstadt.de; i.schroeder@gsi.de

model to study the impact that radiation has on stem and progenitor cells within the neuronal network. As these cells mainly govern the maintenance and regeneration of the normal neuronal tissue, their contribution to the radiation-induced CEL is of great importance to avoid or treat such potential debilitating side effects. While CELs often only represent late stage RN and biopsy material is scarce and only from severe cases, cerebral organoids offer the unique opportunity to study the etiology and progression of the normal tissue responses which lead to CEL. The cerebral organoid model has been used with great success to uncover fundamental mechanisms and alterations in normal brain tissue for various neurological diseases, serving as a bridge between knowledge gained from basic in vitro-, animal- and human epidemiological studies[20]. Since CEL are of particular concern for pediatric patients, who demonstrate high cure rates and subsequently expected long lifespans, we used high-energy protons, which are preferably used for the treatment of pediatric malignancies, in addition to high-energy X-rays, which are received by the majority of cancer patients, in our study.

## Results

As CELs occur in various brain regions depending on the localization of the tumor and the radiation field, we used an unguided differentiation approach according to Lancaster et al. [21,22]. (Fig. 1) to generate cerebral organoids in which neural progenitors are allowed to differentiate spontaneously with maximum diversity. As shown in Fig. 1, organoids were irradiated after 20 days (d) of maturation at which time they are mainly comprised of sex-determining region Y (SRY)-box 2 (SOX2)[+]/nestin (NES)[+]/paired box protein-6 (PAX6)[+] neuroepithelial progenitor cells clustered in evolving ventricular zone-like rosettes (immature organoids, Supplementary Fig. 1a, b[20,23],). Organoids were also irradiated at d80 when the neuronal network matured. In contrast to d20 organoids, d80 organoids expressed the neuronal marker microtubule-associated protein 2 (MAP2) adjacent to regions with PAX6[+] neuronal progenitors (mature organoids, Supplementary Fig. 1a, b). Given that d20 organoids still differentiate substantially, changing their cellular composition accordingly, samples were taken approximately 40 days later (at d60) to allow for the observation of radiation impacts on cell differentiation. In contrast, the more mature organoids, where no dramatic changes in cell differentiation and thus cell composition occur, were irradiated at d80 and followed up to d140 even though most analyses were performed at d100.

### Exposure to radiation results in growth retardation and reduced proliferation, but is not correlated to necrosis

Cerebral organoids, generated with current protocols, represent the prenatal embryonic/fetal or early postnatal brain, respectively. Prenatal radiation exposure leads to growth retardation and small head/brain size (reviewed in B. Yang et al. [24]). Thus, we hypothesized the d20 organoids, representing a prenatal brain, to be very sensitive to irradiation responding with growth

retardation reflected in a smaller organoid size while the d80 organoids were expected to be less radiation-sensitive. To study the impact of radiation, cerebral organoids were subjected to X-ray and proton irradiation (Fig. 1). X-rays were used as reference radiation and all samples were compared to sham-irradiated controls (0 Gy). Due to the above-mentioned fetal/neonatal radiosensitivity, originating from neuroepithelial stem cells[25], for immature organoids (d20) doses of 1, 2, and 8 Gy were used. For the more mature organoids (d80), 3, 10, and 15 Gy were applied. In addition, d80 organoids were irradiated with high-energy protons (3, 10, and 15 Gy), both in the plateau (normal tissue) phase and in the Spread Out Bragg Peak (SOBP; tumor tissue), commonly used for radiotherapy of pediatric cancers[26].

Growth was evaluated via commonly used microscopic bright field imaging (area measurement[27], Fig. 2a, b) and evaluated in conjunction with cell proliferation (Ki67 staining, Fig. 2c, d), necrosis (LDH release assay, Fig. 2e, f) and apoptosis (Caspase 3-active and 3/7 assays, Supplementary Fig. S2). Measurements of the circular area (Fig. 2a, b) revealed that compared to the controls at the day of irradiation (dotted lines), all sham controls showed an increase in size (~84% increase for d60 organoids compared to the size at d20 and ~56% increase for d100 organoids compared to the size at d80). Organoids irradiated at d20 with X-rays (Fig. 2a) displayed a significant dose-dependent decrease in size at d60 when compared to the respective sham-controls (~31% decrease after 1 Gy, ~52% decrease after 2 Gy and ~92% decrease in size after 8 Gy). In the highest dose-cohort (8 Gy), this decrease resulted in an even smaller size than the d20 samples (analyzed 40 d prior) exhibited. The response of mature organoids irradiated on d80 (Fig. 2b) was less drastic. Here significant growth retardation was only observed in samples subjected to 10 Gy (~20% decrease in size) and 15 Gy X-rays (~28% decrease in size) or 10 Gy (~39% decrease in size) and 15 Gy proton irradiation in SOBP (~43% decrease in size), corresponding to the irradiation of the tumor bed. Irradiation in the entrance channel (plateau, corresponding to the surrounding normal tissue) was not significantly altered compared to the sham controls. This irradiation-related decrease in size corresponded to a decrease in proliferation as determined by Ki67 staining (Fig. 2c, d). While d20 organoids showed ample Ki67 staining throughout the neuronal rosettes/proliferation zones prior to irradiation, Ki67 was restricted to the inside of some, but not all neuronal rosettes in d60 sham irradiated controls (Fig. 2c). Irradiation with 1 Gy X-rays led to a decrease in proliferation, while Ki67 was not detectable anymore in the mostly deteriorated organoids subjected to 8 Gy X-rays. Due to their more mature character/progressed differentiation stage, d80 organoids and their d100 counterparts showed hardly any proliferation (Fig. 2d). However, irradiation with 3 and 15 Gy X-rays further diminished Ki67 expression.

For the measurement of necrosis, the lactate dehydrogenase (LDH) assay was used, and LDH release was measured on d21, d40, and d60 for immature organoids (corresponding to 1 d, 20 d, and 40 d after irradiation), and on d81, d100, and d140 for mature organoids (corresponding to 1 d,

**Fig. 1 | Differentiation scheme and characteristics of organoids at d20 (immature) and d80 (mature).** An Experimental scheme (created with BioRender.com).

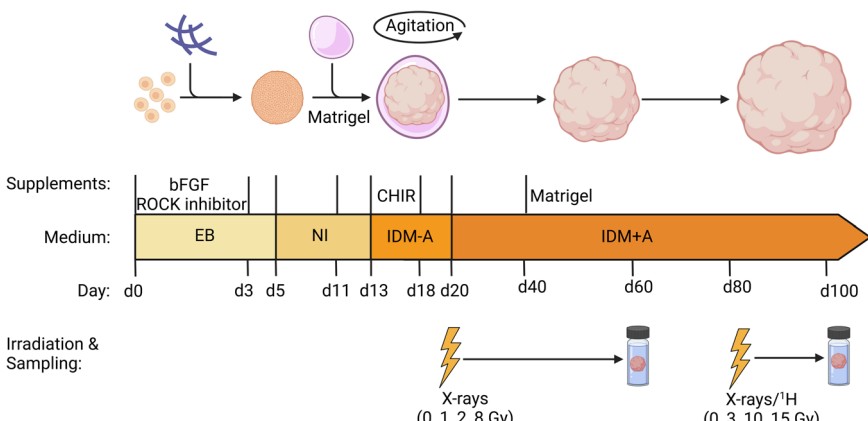

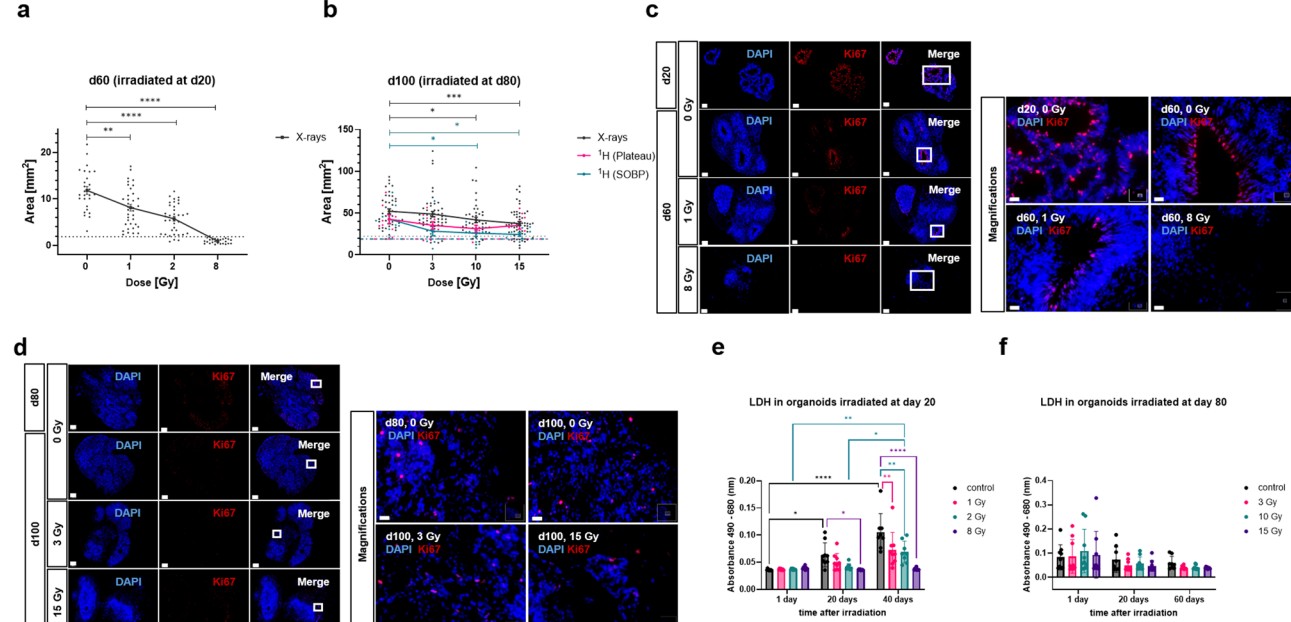

**Fig. 2 | Size measurement and analysis of necrosis in organoids. a, b** The organoid size measured as circular area [mm²] for organoids irradiated at d20 (immature) and at d80 (mature) measured 40 and 20 days post-irradiation (d60 and d100), respectively, using different radiation qualities and doses: X-rays: 1–8 Gy for d20 organoids and 3–15 Gy for d80 organoids; protons in SOBP or plateau, respectively: 3–15 Gy for d80 organoids in comparison to their respective sham-irradiated controls. Dashed lines delineate the size of the controls at the day of irradiation. Statistical analysis was done either by Brown-Forsythe and Welch ANOVA or one-way ANOVA with Dunnett´s post-test. Data are presented as mean ± SEM for three independent experiments ($N = 3$) and 25 to 36 organoids per experiment ($n = 25–36$) (X-rays, d20), $N = 4$ and $n = 40-54$ (X-rays, d80), $N = 1$ and $n = 6–9$ (protons in plateau, d80) and $N = 1$ and $n = 9–10$ (protons in SOBP, d80). **$p = 0.0039$; mean difference ± SE of difference: 3.704 ± 1.092; 95% CI of difference: 1.017 to 6.39 for 1 Gy at d20; ****$p < 0.0001$; mean difference ± SE of difference: 6.094 ± 0.9881; 95% CI of difference: 3.647 to 8.540 for 2 Gy at d20; ****$p < 0.0001$; mean difference ± SE of difference: 10.90 ± 0.8476; 95% CI of difference: 8.750 to 13.05 for 8 Gy at d20; *$p = 0.0448$; mean difference ± SE of difference: 10.35 ± 4.299; 95% CI of difference: 0.1884 to 20.52 for 10 Gy X-rays at d80; ***$p = 0.0008$; mean difference ± SE of difference: 14.78 ± 4; 95% CI of difference: 5.320 to 24.23 for 15 X-rays Gy at d80; *$p = 0.0360$; mean difference ± SE of difference: 16.75 ± 6.431; 95% CI of difference: 0.9284 to 32.57 for 10 Gy protons at d80; *$p = 0.0209$; mean difference ± SE of difference: 18.22 ± 6.431; 95% CI of difference: 2.396 to 34.04 for

15 Gy protons at d80. **c,d** Representative immunofluorescence staining of Ki67 (red) in d20 and d80 cerebral organoids, nuclei stained with DAPI (blue), scale bar: 200 µm, and 20 µm for magnification. **e, f** Lactate dehydrogenase (LDH) levels secreted by organoids irradiated with X-rays at d20 (0–8 Gy), measured 1, 20 and 40 days post-irradiation, and secreted by organoids irradiated at d80 (0–15 Gy), measured 1, 20 and 60 days post-irradiation. LDH level is directly proportional to absorbance values measured at 490 nm and subtracted from background values measured at 680 nm. Statistical analysis was done by two-way ANOVA with Tukey´s post-test. Data are presented as mean ± SD for three independent experiments ($N = 3$) and three organoids per experiment ($n = 3$). *$p = 0.0263$; predicted mean difference: −0.02798; 95% CI of difference: −0.05424 to −0.001730 for 1 day control vs. 20 days control; ****$p < 0.0001$; predicted mean difference: −0.06953; 95% CI of difference: −0.09659 to −0.04247 for 1-day control vs. 40 days control; **$p = 0.0080$; predicted mean difference: −0.03187; 95% CI of difference: −0.05893 to -0.004810 for 1 day vs. 40 days (2 Gy); *$p = 0.0303$; predicted mean difference: 0.02762; 95% CI of difference: 0.001363 to 0.05387 for control vs. 8 Gy (20 days); *$p = 0.0443$; predicted mean difference: −0.02740; 95% CI of difference: -0.05447 to -0.0003435 for 20 days vs. 40 days (2 Gy); **$p = 0.0048$; predicted mean difference: 0.03308; 95% CI of difference: 0.006016 to 0.06014 for control vs. 1 Gy (40 days); **$p = 0.0017$; predicted mean difference: 0.03646; 95% CI of difference: 0.008610 to 0.06430 for control vs 2 Gy (d40); ****$p < 0.0001$; predicted mean difference: 0.06822; 95% CI of difference: 0.04116 to 0.09528 for control vs 8 Gy (d40).

20 d, and 60 d after irradiation) (Fig. 2e, f). In contrast to the general hypothesis that irradiation leads to high levels of necrosis, a significant increase in extracellular LDH release was only observed in the sham controls from 1 d to 40 d post-irradiation and a slight but not significant increase in the samples exposed to 8 Gy X-rays 1 d after irradiation (Fig. 2e). At all other time points, LDH release was lower in irradiated organoids compared to the respective sham controls. The most significant decrease was observed in organoids 40 d post-irradiation while in samples irradiated at d80, no significant changes could be observed at any time point (Fig. 2e, f).

To elucidate any apoptotic events, caspase staining was performed (Supplementary Fig. S2). 1 d after X-ray irradiation, organoids revealed a dose-dependent increase of apoptotic cells that was more pronounced in d20 organoids compared to d80 organoids (Supplementary Fig. S2a, b). However, at later time points (d60 or d100, Supplementary Fig. S2c, d), this response was not observed.

**Irradiation leads to the formation of liquid-filled cavities**

Irradiation of cerebral organoids resulted in the formation of liquid-filled cavities with a typical diameter between 0.1–0.8 mm (Fig. 3). However,

extreme cases of these cavities were observed with one becoming 18.7 mm in diameter with the corresponding organoid having a diameter of 2.6 mm (Fig. 3a). The occurrence of these cavities, depicted in Fig. 3b, c, was as dose-dependent as the growth retardation of organoids (Fig. 2a, b), and it was highest in organoids irradiated with 8 Gy X-rays at d20 (Fig. 3b). In organoids irradiated at d80 (Fig. 3c), protons in the SOBP were more effective in inducing cavity formation than X-rays or protons in the plateau region at the same dose. The slope of the fitted dose-response curves (Supplementary Fig. S3) and the relative biological effectiveness (RBE) of proton beams compared to X-rays are reported in Supplementary Table 1.

**Radiation-induced liquid-filled cavities represent choroid plexus**

Liquid-filled cavities observed after irradiation of cerebral organoids morphologically resembled choroid plexus with cerebrospinal fluid (CSF)-filled cavities[28,29]. The choroid plexus (CP) is comprised of highly vascularized stroma with fenestrated capillaries and epithelial tissue, which forms the blood-CSF barrier and is the production site for CSF in the vertebrate brain[30]. The CP regulates the entry of compounds into the brain, its development, and function. The CP epithelium arises from multipotent pre-

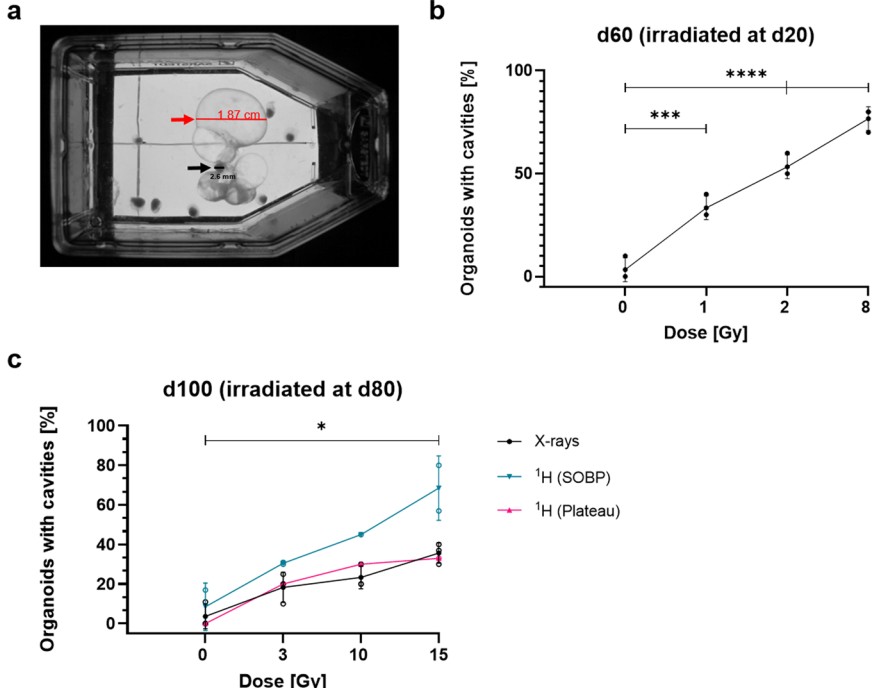

**Fig. 3 | Cavity formation in cerebral organoids after irradiation with various radiation qualities. a** Organoid, irradiated at d80 with 15 Gy protons (Plateau region), displays massive liquid filled cavities. **b** Percentage of organoids displaying cavities after irradiation with X-rays at d20 (0–8 Gy) and **c** with various radiation qualities: X-rays, protons in SOBP and in plateau, respectively at d80 (0–15 Gy). Quantitative analysis of the dose-response curve including a linear fit of the data and relative biological effectiveness are reported in Supplementary Table 1 and Supplementary Fig. S3. Statistical analysis was done by one-way ANOVA with Dunnett´s post-test for organoids at d60 and d100 where applicable. Data in **b-c** are presented as mean ± SD for two to three independent experiments ($N = 2$–3) or as single experiment ($N = 1$) and $n = 10$ organoids per variant/group. ***$p = 0.0006$; mean difference ± SE of difference: $-30 \pm 4.714$; 95% CI of difference: $-43.58$ to $-16.42$ for 1 Gy at d20; ***$p < 0.0001$; mean difference ± SE of difference: $-50 \pm 4.714$; 95% CI of difference: $-63.58$ to $-36.42$ for 2 Gy at d20; ***$p < 0.0001$; mean difference ± SE of difference: $-73.33 \pm 4.714$; 95% CI of difference: $-86.91$ to $-59.76$ for 8 Gy at d20; *$p = 0.0127$; mean difference ± SE of difference: $-19.67 \pm 5.137$; 95% CI of difference: $-34.46$ to $-4.873$ for 10 Gy at d80; ***$p = 0.0007$; mean difference ± SE of difference: $-32 \pm 5.137$; 95% CI of difference: $-46.79$ to $-17.21$ for 15 Gy at d80.

neurogenic neuroepithelial stem cells[31], rather than later-stage neurogenic radial glia, and the CP in turn supports these stem cells[30]. In addition, CP formation is preceded by and depends on cortical hem (CH) signaling by BMPs and WNTs (reviewed in Moore and Iulianella[32]). Therefore, we examined the expression of neuroepithelial stem cell, CH and CP markers in organoids before and after irradiation as shown in Figs. 4 and 5 and Supplementary Figs. S4 and S6.

qPCR analyses revealed that irradiation at d20 did not significantly change the expression of *SOX2* mRNA in 60 days old organoids compared to the sham-irradiated control (Fig. 4a). A slight but not significant decrease in the expression of *PAX6* and *MAP2* mRNA was observed after irradiation with 1 Gy X-rays only, whereas irradiation of organoids with the dose of 8 Gy X-rays resulted in a significant increase of *NES*, and a slight but not significant increase in *PAX6* and *MAP2* mRNA level compared to the sham irradiated controls (Fig. 4a). Exemplary d60 IF analyses of sham-irradiated controls displayed neuronal rosettes whose cores were predominantly NES[+]/PAX6[+] with scarce, scattered SOX2 expression surrounded by MAP2[+] cells (Fig. 4b, c). Cells positive for homeodomain-only protein (HOPX), which is known as a marker of outer radial glial cells (oRGs)[33], were particularly prominent at the rim, and GFAP[+] cells were scarcely present (Supplementary Fig. 4a). In contrast, organoids subjected to 1 Gy X-rays (Fig. 4b, c, middle panel) showed NES[+]/PAX6[+] rosette-like structures surrounded by MAP2[+] cells as seen in the controls, but they also displayed highly SOX2[+]/PAX6[-] epithelial-like structures that were partially NES[+] and bordered cavities. In general, SOX2 expression was more pronounced compared to the sham controls. HOPX and GFAP showed a slight decrease or no expression change in organoids subjected to 1 Gy X-rays compared to their respective sham controls (Supplementary Fig. 4a). Organoids irradiated with the highest dose at d20 showed deterioration,

which made their evaluation more difficult (Fig. 4b, c, Supplementary Fig. S4a, b). Nevertheless, irradiation with 8 Gy X-rays led to areas with prominent nuclei condensation, identified by DAPI staining. MAP2 staining was restricted to these areas, whereas adjacent apparently vital cells stained for SOX2 and partially for PAX6 and NES. HOPX staining was generally less intense and observed throughout the entire organoid, while GFAP[+] cells were mostly absent when compared to their respective controls (Supplementary Fig. S4a).

Irradiation at d80 did not significantly change the mRNA expression of *SOX2* and *MAP2* in organoids at d100 compared to the sham-irradiated control (Fig. 4d). A slight, but not significant increase in the expression of *NES* mRNA level was observed. *PAX6* mRNA level was slightly but not significantly increased after 3 Gy X-rays, but slightly although not significantly decreased after 15 Gy X-rays compared to sham irradiated controls (Fig. 4d). Exemplary IF analyses show structural changes in irradiated compared to non-irradiated sham controls in Fig. 4e, f, for instance, sham controls at d100 were devoid of rosette-like structures. However, cells with low SOX2 expression were found throughout the organoid, while PAX6[+] cells were predominantly detectable as a rim adjacent to areas with higher MAP2 expression or intermingled with cells displaying lower MAP2, SOX2, and NES expression (Fig. 4e, f). The rim of the organoid was enriched with HOPX[+] cells, and less dominant astrocytes with an intense expression of GFAP (Supplementary Fig. 4b). Irradiation with 3 Gy X-rays (Fig. 4e, f, Supplementary Fig. S4b, middle panel) led to scant SOX2 expression. Even though NES expression was detected as well, co-staining of both markers was rarely detected. MAP2 was strongly expressed at the outer rim of structures that resembled late rosette-like features, but these were devoid of PAX6. HOPX and GFAP both showed a slight decrease in the expression compared to their respective sham controls. Irradiation with 15 Gy X-rays

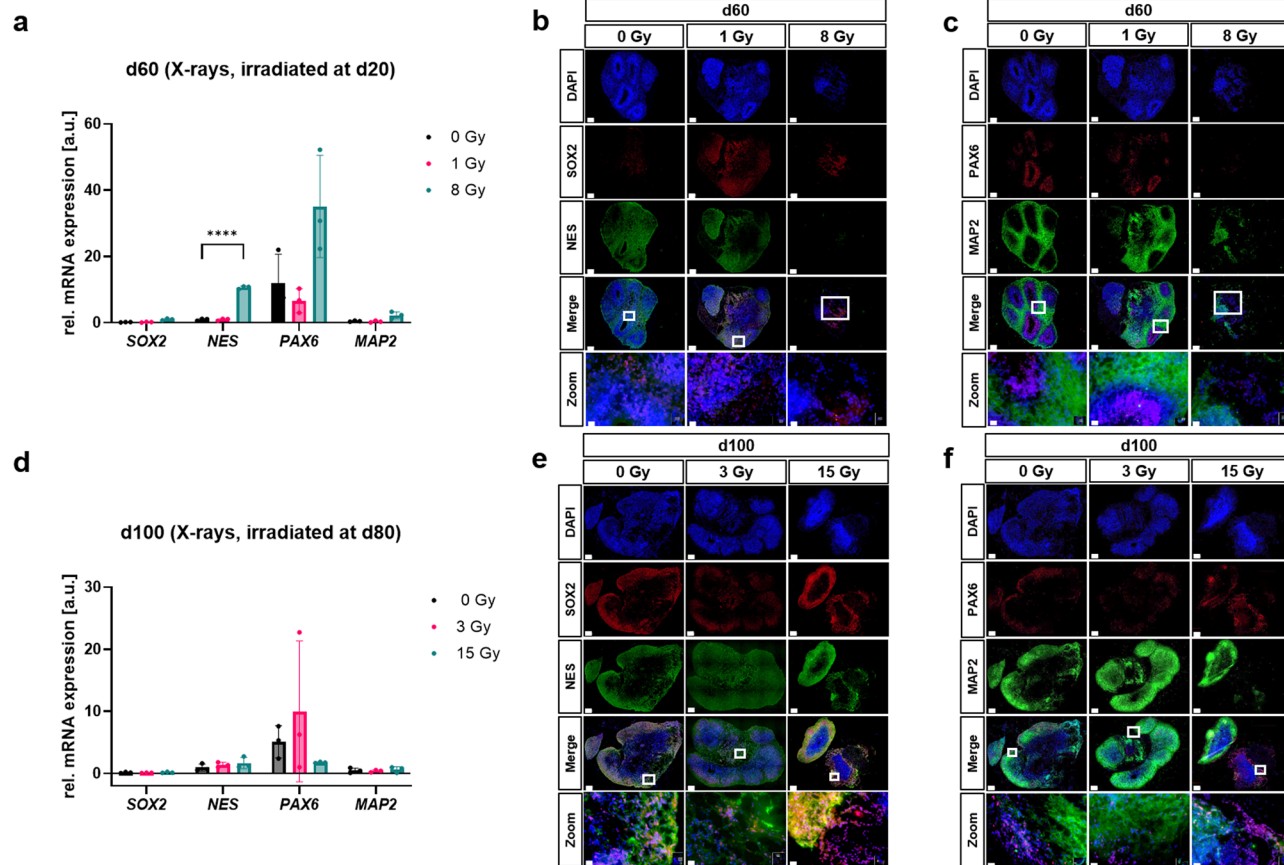

**Fig. 4 | Expression of markers for neural stem and progenitor cells and mature neurons in irradiated organoids and their sham controls. a** Relative mRNA expression of *SOX2*, *nestin* (*NES*), *PAX6*, and *MAP2* in d66 organoids subjected to 8 Gy X-rays at d20, mean ± SD for three independent experiments (*N* = 3) and *n* = 3 organoids per experiment. Values were normalized to 18S rRNA levels. Statistical analysis was done by multiple unpaired two-tailed *t*-test or Welch´s *t*-test, \*\*\*\**p* = 0.000016 for*;* effect size ± SE of difference: 9.694 ± 0.2900 for *NES*. **b** Representative immunofluorescence staining of SOX2 (red) and NES (green) or **c** PAX6 (red) and MAP2 (green) at d66 (irradiation at d20). Nuclei stained with

DAPI (blue), scale bar: 200 μm, and d20 μm for magnification. **d** Relative mRNA expression of *SOX2*, *NES*, *PAX6*, and *MAP2* in d100 organoids subjected to 15 Gy X-rays at d80, mean ± SD for three independent experiments (*N* = 3) and *n* = 3 organoids per experiment. Values were normalized to 18S rRNA levels. Statistical analysis was done by multiple unpaired two-tailed *t*-test or Welch´s *t*-test. **e** Representative immunofluorescence staining of SOX2 (red) and NES (green) or **f** PAX6 (red) and MAP2 (green) at d100 (irradiation at d80). Nuclei stained with DAPI (blue), scale bar: 200 μm, 20 μm for magnification.

(Fig. 4e, f; two different organoids) led to a strong SOX2 and NES expression. In some areas, theses markers were co-expressed, but there were also areas with single expression. PAX6/MAP2 co-staining revealed the heterogeneity of organoids: Of the two organoids stained, one was exclusively PAX6+ while the other was almost exclusively MAP2+ (Fig. 4f). HOPX expression was overall less intense compared to the control but this change was not significant. The expression level of GFAP varied strongly between organoids of the irradiated group, but GFAP expression was similar to sham controls and remained restricted to the rim (Supplementary Fig. S4b).

As irradiated organoids apparently retained neuroepithelial stem cells in rosettes and the observed cavities showed similarities to CP structures, we probed for the CH marker MSX1, which instructs the differentiation of the CP. In addition, CH and CP lineage marker LMX1A[34], choroid epithelium marker OTX2[35], and common CP markers zona occludens (ZO) 1, aquaporin (AQP) 1, and claudin (CLDN) 3 were analyzed. We also determined the mRNA level of insulin-like growth factor (IGF) 2 in the irradiated organoids. This marker was found to be highly enriched in CSF and is expressed during brain development[36]. IGF2 was shown to be restricted to a subtype of the CP epithelium, termed dark cells, which were uncovered using CP organoids[37]. The inwardly rectifying potassium channel KIR7.1, which is highly expressed in CP epithelium[38], was chosen as it is a prerequisite in the proper generation of CSF[39].

Compared to sham-irradiated controls, the level of *MSX1* mRNA was significantly increased in samples subjected to 8 Gy X-rays at d20 and analyzed at d60. The levels of *LMX1A*, *OTX2*, *ZO1*, *AQP1*, *CLDN3*, and *IGF2* mRNA, although not significantly upregulated, all showed a slight increase compared to the sham-irradiated control. Only *KIR7.1* was expressed to a lesser extent in the irradiated versus the non-irradiated young organoids (Fig. 5a). A similar expression pattern for all markers could be seen in the d100 organoids that were X-ray-irradiated at d80 (Fig. 5b). Proton irradiation led to substantially higher mRNA expression of *MSX1*, *LMX1A*, *OTX2*, *AQP1*, *CLDN3*, *IGF2*, and *KIR7.1* (Fig. 5c). However, the changes were not significant due to the lower sample size.

The existence of CP-like structures with their distinct morphology as a monolayer of epithelial cells in irradiated organoids was confirmed in young organoids subjected to X-rays at d20 and analyzed at d60 using H&E staining (Supplementary Fig. S5) and immunofluorescence staining against AQP1 and CLDN3 (Fig. 5d, e), whose co-expression is only found in CP. While AQP1 and CLDN3 staining in cells resembling early CP could be observed occasionally in sham controls, irradiated organoids showed multiple AQP1 and CLDN3 lined cavities of various sizes. The same was observed in organoids irradiated at d80 and analyzed at d140. Here, the effects were less pronounced compared to the d20 organoids reflecting the same trend seen in the qPCR analyses. The CP-like structures were also positive for the tight junction protein ZO1 and CLDN3. However, whereas

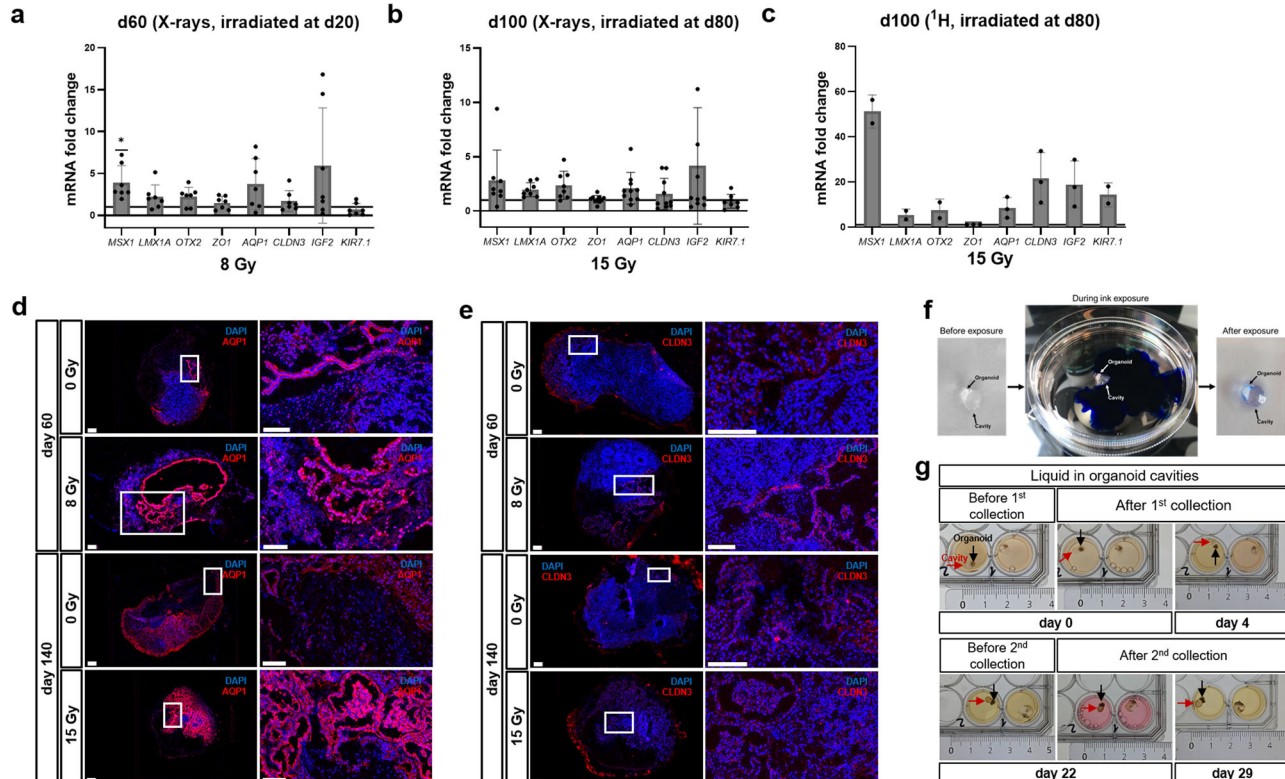

**Fig. 5 | Choroid plexus (CP) formation in irradiated cerebral organoids. a–c** Relative mRNA expression of the CH and CP lineage marker *MSX1* and *LMX1A*, choroid epithelium marker *OTX2*, CP markers *ZO1*, *AQP1*, *CLDN3*, *IGF2* and *KIR7.1* in d66 organoids subjected to 8 Gy X-rays at d20, mean ± SD for at least four independent experiments (*N* = 4) and *n* = 3 organoids per experiment, and in d100 organoids subjected to either 15 Gy X-rays or 15 Gy protons at d80, mean ± SD for at least five independent experiments (*N* = 5) and *n* = 3 organoids for X-rays and for two to three independent experiments (*N* = 2–3) and *n* = 3 organoids for protons.

Values were normalized to 18S rRNA levels and expressed relative to values for sham-irradiated control. Statistical analysis was done by two-tailed Welch´s *t*-test, \**p* = 0.0208; effect size ± SEM: 0.4363 ± 0.1454; 95% CI: 0.09006 to 0.7826 for *MSX1*. The lines indicate the levels in the sham-irradiated controls. **d, e** Representative immunofluorescence stainings of the CP markers AQP1 (red) and CLDN3 (red), and nuclei stained with DAPI (blue), scale bar = 100 μm, **f** Cerebral organoid after ink-based barrier test. **g** Cerebral organoids before and after collection of the fluid from organoid cavities.

the fluorescence signal for CLDN3 was restricted to CP-like structures, ZO1 was found in both, ventricular/neuroepithelial proliferation zones and CP-like structures (Supplementary Fig. S6). The barrier function of the cell layer lining the cavities was directly assessed by examining the entry of ink from the medium into the cavities. The ink was completely excluded from the cavities, proving intact barrier characteristics over the observed time of up to 5 min (Fig. 5f). The fluid contained within the cavities was actively produced by organoids as the contents were completely restored within a few days after its removal from the cavities (Fig. 5g).

**Altered WNT/BMP signaling reflects organoid's response to irradiation and CP formation**

Irradiation to the brain can affect neurogenesis[40,41] by inducing changes in genes that regulate proliferation, cell cycle, and differentiation[42]. Signaling pathways that govern neurogenesis and/or choroid plexus formation include neurogenic locus notch homolog protein (NOTCH)[43], wingless-related integration site (WNT)[44,45] and bone morphogenic protein (BMP) signaling[46], NOTCH directly induces hairy and enhancer of split (HES) 1, 3, and 5, whose expression leads to the specification of CP epithelium at the expense of neurogenin (NGN) 2. Thus, we assessed the mRNA expression of *NOTCH1* and *2*, *NGN2* as well as *HES1* and *5* (Fig. 6). Compared to the sham controls, *NOTCH1* and *2*, and *NGN2* showed similar expression, while *HES5* showed a slight increase and *HES1* was significantly increased in organoids irradiated at d20 (Fig. 6a). In organoids exposed to X-ray or proton irradiation (SOBP) at d80, the mRNA expression of *NOTCH1* and *2*, *NGN2*, *HES1* and *HES5* were slightly decreased compared to the respective sham controls (Fig. 6b, c).

The WNT signaling pathway is known to also play a role in CP formation. WNT3a null mutants exhibit smaller CP areas, while treatment with WNT3 supports BMP4-mediated CP derivation from embryonic stem cells[35]. WNT5a promotes the formation of the tight epithelium in the developing CP and lack of WNT5a is associated with reduced CP size and complexity and loss of apicobasally polarized morphology[44]. As WNT signaling via b-catenin acts as a co-activator of lymphoid enhancer factor 1(LEF1)/TCF transcription factors[47], we included LEF1 in the analyses. mRNA levels of *WNT3*, *WNT5a*, and *LEF1* were elevated in both, organoids irradiated at d20 and d80 irrespective of the radiation quality (Fig. 6). These changes were significant for *WNT5a* and *LEF1* in the d20 irradiated samples (Fig. 6a). *BMP4*, which is regulated by *WNT3a* (27) and is sufficient for CP epithelium induction[46], was upregulated in organoids subjected to X-rays at d20 and after proton irradiation at d80 albeit with higher variability, while no increase was observed after d80 X-ray irradiation (Fig. 6b, c).

## Discussion

Cerebral organoids have been used to mimic the onset and progression of several neurological diseases[20] including the dysregulation of brain and CP cell types in patients with severe SARS-CoV-2 infections[28,29]. Here, we use cerebral organoids to elucidate the nature of CELs that are generally attributed to radiation necrosis as a result of damage to the vasculature of the brain and the blood-brain barrier. In contrast to the general opinion that radiation induces necrosis, irradiated cerebral organoids generated via unguided differentiation displayed lower levels of necrosis than sham-irradiated controls. This effect - most striking in d20 samples analyzed at the latest time point - stems from an intrinsic issue of the organoid model

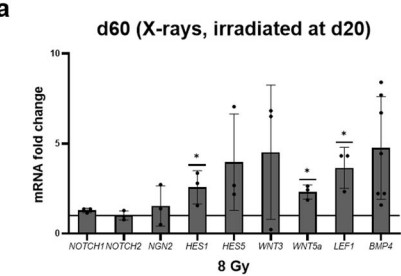
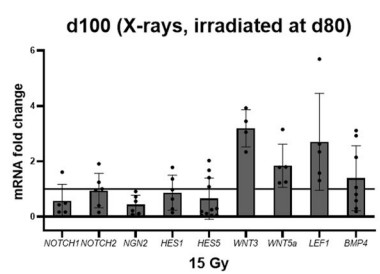
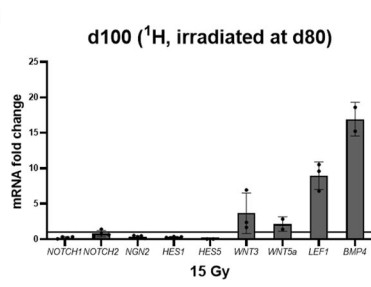

**Fig. 6 | NOTCH/HES and WNT/BMP signaling in irradiated cerebral organoids.**
**a–c** Relative mRNA expression of the markers *NOTCH1*, *NOTCH2*, *NGN2*, *HES1*,
*HES5*, *WNT3*, *WNT5a*, *LEF1* and *BMP4* in d66 organoids subjected to 8 Gy X-rays at
d20, mean ± SD for at least three independent experiments ($N$ = 3) and $n$ = 3
organoids per experiment, and in d100 organoids subjected to either 15 Gy X-rays,
mean ± SD, for at least four independent experiments ($N$ = 4) and $n$ = 3 organoids
per experiment or to 15 Gy protons (SOBP) at d80, for at least two independent

experiments ($N$ = 2) and $n$ = 3 organoids per experiment. Values were normalized to
*18S rRNA* levels and expressed relative to values for sham-irradiated control. Sta-
tistical analysis was done by two-tailed Welch´s *t*-test, *$p$ = 0.0122; effect size ± SEM:
61.16 ± 14.06; 95% CI: 22.11 to 100.2 for *HES1*; *$p$ = 0.0145; effect size ± SEM:
0.05273 ± 0.01277; 95% CI: 0.01727 to 0.08819 for *WNT5a*, *$p$ = 0.0233; effect size ±
SEM: 0.006565 ± 0.001837; 95% CI: 0.001464 to 0.01167 for *LEF1*. The lines indicate
the levels in the sham controls.

deprived of vasculature: as they enlarge, reaching diameters of several
millimeters, they exceed the limit of passive diffusion of oxygen and nutrient
supply, resulting in necrotic cells at the core[48]. Initial radiation-induced cell
killing, particularly evident by the nuclei condensation and positive staining
for an apoptotic marker after exposure of young organoids to 8 Gy X-rays,
and decrease in cell proliferation led to the observed growth retardation (i.e.
decrease in the size) in organoids, thus, resulting in an alleviated necrosis in
the irradiated organoids compared to the controls.

However, we show that CEL-like appearance involves the blood-CSF
barrier or choroid plexus, and the formation of liquid-filled cavities is clearly
attributable to the radiation impact. Cerebral organoids generated in an
unguided differentiation process according to the protocol by Lancaster et al.
[21,22]. are capable of developing CP. However, this process is rare and occurred
in less than 10% of all control organoids in our study, compared to up to 50-
70% in irradiated organoids. Indeed, to generate CP organoids, dorsalization
has to be extrinsically induced using bone morphogenic protein (BMP)
4[37] that both in vitro and in vivo stimulates epithelial cell formation within
the dorsal midline by repressing forkhead box (FOX) G1, which in turn
supports neural stem/progenitor cell proliferation[49]. The formation of
liquid-filled cavities in such induced conditions is observed after differ-
entiation for more than 28 days and follows the appearance of characteristic
cuboidal epithelium as has been shown in our study as well. Therefore, the
radiation impact mimicked the guided differentiation of CP organoids as
judged also by the expression of CP markers such as AQP1, the water
channel essential for CSF production, and the tight junction proteins ZO1
and CLDN3, which ensure the integrity of the CP[50]. While ZO1, even
though being a more general marker of tight junctions, is found in neu-
roepithelial proliferation zones and CP-like structures, the morphology of
the latter is clearly distinguishable, rendering ZO1 a suitable marker for CP
in addition to the others used in this study. Another characteristic marker is
the potassium channel KIR7.1, although it was reported to some extent in
Purkinje neurons and pyramidal neurons in the hippocampus[51], in glia
cells[52], and retinal pigment epithelium[53] as it is mainly expressed in secretory
cells of the CP[38,54]. A slight increase in the mRNA expression of *KIR7.1* in
response of mature organoids to proton irradiation only, suggests induction
of changes in the function of the CP depending on both the organoid age and
quality of radiation. The existence of members of the NOTCH/HES sig-
naling pathway and especially the induction of WNT/BMP signaling, which
are both required for the regulation of CP formation in vivo[30,35,44], together
with unchanging or decreased neural *NGN2* have been identified as being
the cause for the altered differentiation process by the radiation impact. For
the NOTCH signaling, this effect was more pronounced in the immature
organoids than in the mature ones, while WNT signaling induction was
observed in all organoids irrespective of their maturation status and the type
of irradiation used. This agrees with the findings that using the WNT
activator CHIR in combination with BMB4 is sufficient to induce CP in

cerebral organoids[37]. The expression of WNT ligands implies the presence of
hem-like cells expressing MSX1 that secrete these ligands, which precede the
formation of CP ([34], reviewed in ref. [50]). A significant increase in the
expression of *MSX1* in d60 organoids after irradiation, and a tendency
towards an increase in *MSX1*, lineage markers *LMX1A* and *OTX2* regardless
of the organoid age and irradiation quality, support the gradual differ-
entiation into CP. The enrichment of CP cells in cerebral organoids irra-
diated at d14 was also observed in a recent study[55].

These findings open an alternative interpretation of the CEL seen in
brain tumor patients who received radiation therapy, which is substantiated
by two clinical studies. Recently, Eulitz et al. showed an increased radio-
sensitivity in the periventricular region correlating with an increase in late
radiation-induced brain injuries (RIBI) when evaluating consecutive MRI
scans as a follow-up after proton therapy in patients with glioma[56]. The
median distances of the RIBI volume centers to the cerebral ventricles,
where the CP is located, and to the clinical target volume border were
2.1 mm and 1.3 mm, respectively. The same correlation of CEL formation
and proximity to the periventricular region has been observed in a second
study[57]. From our findings and the onset of RIBI/CEL in the ventricular
region, involvement of the CP in the onset of such lesions is likely. Our study
implies that radiation induces focal CP formation from neuroepithelial stem
cells leading to the typical frond-like structures with increasing CSF pro-
duction that have great resemblance to the CEL seen in MRI scans of
patients who received cerebral radiotherapy. Even though in our short-time
observations, CP barrier function was intact, arguing against a possible
in vivo leakage of contrast-enhancing agents into the CP cavities, a com-
parison of 5 min and 4 h post-intravenous (IV) administration of a single
dose of gadolinium-based contrast agents in a small cohort of patients with
clinically suspected endolymphatic hydrops revealed significant leakage of
contrast agents into the CSF spaces only at 4 h, but not at 5 min post-IV[58].
Thus, a leakage into the newly formed CP cavities is still possible in vitro and
in vivo. In vivo, such leakage has been observed under multiple circum-
stances: in patients with stroke, in those with previous surgery, and those
with high-grade gliomas[59]. The ability of organoids to actively secrete fluid,
found in their cavities, resembles that of CP to secrete CSF. It has been
shown that embryonic as well as adult CSF can support adult neural stem
cells in culture via IGF2[36]. IGF2 levels are attributed largely to CP secretion
and peak during brain development. CP/CSF-IGF2 simulates cell divisions
by binding to receptors on the surface of neural stem cells[36]. In our study,
*IGF2* was found to be upregulated in mature organoids indicating CSF
production and secretion. Likewise, neurogenesis can occur from neural
precursors within the developing choroid plexus[60]. Therefore, the formation
of CP at the expense of newly formed neurons and outer radial glial cells, as
evidenced by the decrease in the expression of MAP2 and HOPX, occurred
as a dose-dependent response of neuroepithelial stem cells (SOX2[+]/NES[+]/
PAX6[+]) to irradiation. The late production of CSF may thus explain the late

onset, but progressive enlargement of CEL, and why pediatric patients are more severely affected than adult patients, as the pediatric brain still exhibits a higher plasticity than the adult brain[61]. In addition, the efficacy of the current approach to treat CEL with bevacizumab does not contradict the involvement of CP formation. Even though this humanized anti-VEGF monoclonal antibody is applied to inhibit vascular growth and to normalize the blood-brain barrier[62], thus alleviating brain damage caused by radiation-induced vascular injury, it will also impact CP cells as VEGF is required for CP maintenance and VEGF receptors are found in adult CP[63]. Likewise, thalidomide has been found to restore the blood-brain barrier and cerebral perfusion in a mouse model of RIBI. This restored function was attributed to the induction of platelet-derived growth factor receptor β (PDGFRβ) expression and subsequent rescue of pericytes[64]. However, thalidomide also downregulates LEF1, the co-activator of the WNT/β-catenin signaling required for CP formation as well as IGF2, which is crucial for CP generation and CSF secretion (see above). Therefore, like bevacizumab, thalidomide has a potentially inhibiting effect on CP formation/progression and CSF production[65]. Assuming that focal/excessive CP/CSF formation is the cause of at least some of the CEL observed in brain tumor patients, additional drugs such as topiramate or the diuretics acetazolamide and furosemide are a treatment option as well, as they decrease secretion of CSF from the CP[66].

## Conclusion

The formation of CP and underlying alterations in the WNT/BMP4 signaling in response to radiation as a cellular stressor has not been previously described. It appears irrespective of the radiation quality but strongly depends on the age of the neuronal tissue that is exposed. CP formation offers an alternative interpretation of the initiation and progression of radiation-induced CEL seen in brain cancer patients and opens up possible new treatment regimes.

## Materials and Methods

### Human embryonic stem cell culture

The feeder-independent hES cell line WA09-FI (H9) was used for all studies presented as approved according to §4 and §6 of the German Stem Cell Act (registry numbers 3.04.02/0125 and 3.04.02/0125-E01). The line was originally generated by the group of Dr. James Thomson at the University of Wisconsin (52). H9 cells were obtained from the WiCell Research Institute, Wisconsin, USA, at passage 23 and were used for experiments in passages 43–53. Cells were routinely cultured on Laminin-521-coated culture dishes (BioLamina, #600962, 10 µg/ml) in mTeSR1 medium (STEMCELL Technologies) supplemented with 50 U/ml penicillin and 5 µg/ml streptomycin (Merck, #A2212) and passaged every 7 d using ReleSR (STEMCELL Technologies, #05872). Briefly, the medium was aspirated, and cells were rinsed first with 2 ml PBS and then with 200 µl ReleSR. After aspiration of the non-enzymatic passaging reagent, cells were incubated for 2 min at 37 °C to detach only pluripotent cells. Detachment was stopped by the addition of pre-warmed mTeSR1 medium and cells were seeded to about $1.0 \times 10^5$ cells per cm$^2$.

### Generation of cerebral organoids

Cerebral organoids were generated as previously described[21,22]. Briefly, for the generation of embryonic bodies (EBs), H9 cells were detached using ReLeSR (Stemcell Technologies) for 3 min at 37 °C. 18000 cells in Embryoid Body medium (EBM) with addition of 4 ng/ml basic fibroblast growth factor (bFGF) and 50 µM Rho-associated protein kinase (ROCK) inhibitor (Tocris Bioscience) were plated into each well of an 96-U-bottom suspension plate (Sarstedt) pre-coated with anti-adherence solution (Stemcell Technologies) for 5 min, 1000 rpm, at RT. The medium was changed to EBM without additional factors on day 3 of culture. EBs were transferred to neural induction (NI) medium in 6-cm dish (Sarstedt) to form neuroepithelial tissues on day 5 of culture. Fresh NI medium was added every second day after transfer. EBs were embedded into Matrigel (Corning) droplets on day 11 and cultured in NI medium until day 13 when it was changed to an improved differentiation medium without vitamin A (IDM-A) with the addition of

3 µM CHIR99021 (Biovision). Medium was then changed every second day until day 18. EBs were then cultured on an orbital shaker in T-25 flasks. The medium was switched to improved differentiation medium with vitamin A (IDM + A) on day 20 of culture, and then changed every 3–4 days. On day 40 of culture, 20 µl/ml Matrigel was added to IDM + A medium.

### Detection of LDH release from necrotic cells

To detect extracellular lactate dehydrogenase (LDH) released from necrotic cells in the media, CyQUANT™ LDH Cytotoxicity Assay Kit (Invitrogen) was used according to the manufacturer's instructions. Briefly, the culture medium was collected at certain time points (1 day, 20 days, and 40 or 60 days) after X-ray irradiation with different doses, i.e. 0 Gy, 1 Gy, 2 Gy, and 8 Gy for organoids irradiated on day 20 of culture or 0 Gy, 3 Gy, 10 Gy and 15 Gy for organoids irradiated on day 80 of culture. On the day of assay, 50 µl of each sample medium, previously stored at -80 °C, was added to a 96-well black clear-bottom plate in duplicate wells. Then, 50 µl of Reaction Mixture from the kit was added to each sample and incubated at room temperature for 30 minutes in the dark to enable conversion of lactate to pyruvate in a reaction catalyzed by LDH where NAD+ is reduced to NADH. The reaction was stopped by adding 50 µl Stop Solution from the kit. The absorbance of red formazan product generated by the reduction of a tetrazolium by diaphorase, which also oxidizes NADH, was measured spectrophotometrically at 490 nm. 680-nm absorbance value (background) measured using SpectraMax i3x plate reader with SoftMaxPro software (v.7.0, Molecular Devices) was subtracted from the 490-nm absorbance. Obtained absorbance values are directly proportional to the amount of LDH released into the media.

### Irradiation of cerebral organoids

At d20 or d80 of differentiation, 5–10 cerebral organoids in T25 suspension flasks were subjected either to X-rays or protons in a dose range of 1–8 Gy (X-rays, d20) or 3–15 Gy (X-rays and protons). X-ray irradiation was performed using a MXR320/26 X-ray tube (250 kV, 16 mA). Exposure to protons took place at the Heidelberg Ion Beam Therapy Center. Samples were subjected either to ion beams in a 30 mm Spread Out Bragg Peak (SOBP, LET: 2.5–8.9 keV/µm) representing the irradiation field of the tumor plus margin or to irradiation conditions in the plateau phase (LET: 0.9–1.5 keV/µm) representing the surrounding normal tissue. Controls were sham-irradiated. The medium was exchanged immediately after irradiation or up to 1 h thereafter.

### Size measurements and quantification of cavity formation

For the measurement of the organoid size, pictures of 10 organoids were taken at d20 and d80 (time of irradiation) and d60 and d100 (40 and 20 d post-irradiation, respectively) using a standard digital camera (Sony DSC-W220). The organoid area was measured using ImageJ. The number of organoids with visible cavities was counted and presented as a percentage of the overall number of organoids.

### Analyses of cavity permeability

To examine the barrier function, an ink test was carried out. For this purpose, 6 drops of dark blue writing ink were carefully added to the cultivation medium under the microscope to observe the permeability of the cavity walls. Videos were recorded for 40 sec at a frame rate of 31.68 frames per second using the Eclipse Ts2 microscope (Nikon). Maximum observation time was ≤ 5 min.

### Analysis of fluid production by CP-like cells in cerebral organoids

To examine whether organoids actively produce fluid leading to the observed cavities, organoids were washed once in PBS and the whole liquid content was collected from cavities using a 1 ml syringe with a 25 G needle and stored at −80 °C. Organoids were placed back into the culture medium in the incubator and monitored daily for the production of liquid in the cavities. Liquid collection from cavities was repeated with the same organoids once the cavities were filled again.

## Immunofluorescence

For immunofluorescence staining, organoids were fixed in 3.7% paraformaldehyde (Carl Roth) at 4 °C overnight and then washed three times for 5 min with PBS. Organoids were dehydrated in sucrose (Sigma) gradient (7–60% sucrose in PBS) for 4 hours (for 7%, 10%, and 40% sucrose) or overnight (for 30% and 60% sucrose). Organoids were then embedded in 7.5% gelatin (Neolab)/10% sucrose using in-house 3D-printed PDMS embedding molds. Embedded organoids were frozen on dry ice and stored at −80 °C prior to being cryosectioned at 10 µm using CM1860 cryostat (Leica Biosystems). Cryosections were first blocked and permeabilized in 0.5% Triton X-100 (ThermoFisher Scientific)/1% BSA (Carl Roth) in PBS for 30 min, and then blocked with 1% BSA in PBS for another 30 min at room temperature (RT). Cryosections were incubated with primary antibodies in 1% BSA in PBS for 1 h at RT or at 4 °C overnight. After being washed three times for 5 min with PBS, cryosections were incubated with secondary antibodies in 1% BSA for 1 h at RT followed by incubation with 5 µg/ml DAPI for 4 min for nuclei staining. Then the cryosections were washed two times for 5 min in PBS followed by two wash steps for 5 min with Millipore water prior to mounting with fluorescence mounting medium (Dako). Images were captured with a fluorescence microscope (Zeiss Axio Imager.Z2) equipped with Metafer5 software (v. 4.3.12, Metasystems). Stainings were performed on samples from three independent preparations with at least three organoids per group. Images were processed with ImageJ (v1.53i, National Institute of Health (NIH)).

Antibodies used for immunofluorescence are listed in Supplementary Tables 2-4.

## Real time RT-PCR analysis

For each condition to be analyzed 3-5 organoids were collected in QIAzol Lysis Reagent (Qiagen, #79306) and total RNA was isolated using the Qiagen RNeasy Mini Kit (#74106) according to the manufacturer´s instructions including a DNA removal step using the RNase-free DNase Set (Qiagen, #79254). 50 ng or 2 µg RNA were reverse-transcribed via the RevertAid RT Kit (Life Technologies, #K1691). Relative RNA expression was analyzed using the Hot FIREPol EvaGreen qPCR Mix Plus from Solis Biodyne (08-24-0000S) and the Quant Studio 3 Real-Time PCR System (Applied Biosystems) with Quant Studio Design & Analysis software (v. 1.5.3). Human fetal and adult brain mRNA served as controls, all target expression levels were normalized to 18S rRNA. Primer sequences can be found in the Supplement Table 5.

## Statistics and reproducibility

Statistical comparisons were performed using GraphPad Prism (v 9.3.1) as stated in figure legends for a number of independent experiments (N) with a given number of organoids per experiment (n) and included an unpaired two-tailed test with or without Welch correction, Brown-Forsythe/Welch or mixed ANOVA model either with Dunnett´s or Tukey´s post-test. Samples of organoids were randomly assigned to different treatments. No statistical methods were used to pre-determine sample sizes. Because of the nature of the treatment (irradiation), data collection and analysis were not performed blind to the conditions of the experiments.

## Reporting summary

Further information on research design is available in the Nature Portfolio Reporting Summary linked to this article.

## Data availability

All data needed to evaluate the conclusions in the paper are present in the paper and/or the Supplementary Materials. Raw data is uploaded on open access repository Zenodo (Version v1, https://doi.org/10.5281/zenodo.14750678)[67]. All other data, such as microscopy images, are available from the corresponding author on reasonable request.

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

## Acknowledgements

The authors thank Kim Sara Jung, Dr. Carola Hartel, Leonie Hartig, and Jennifer Persigehl for excellent technical assistance. We further thank Leon Kaysan and Dr. Timo Smit for microscopic analyses, Dr. Margot Mayer, Dr. M. Waleed Gaber, and Dr. Christine Beamish who shared their time and insights. We are grateful to Dr. Stephan Brons for his help with the planning and performance of proton irradiation experiments at Heidelberg Ion Beam Therapy Center (HIT). This work was funded by US NIH Grant 1RO1CA256848-01, and German Federal Ministry of Education and Research (BMBF) Grant 02 NUK 049A and 02NUK081A. This paper includes parts of results (Fig. 5f, Figure S2 (graphs), Figure S4) of the doctoral thesis "Effects of ionizing radiation on human cerebral organoids" submitted by Celine Schielke to the Johannes Gutenberg University Mainz in 2021 (https://doi.org/10.25358/openscience-6858).

## Author contributions

Conceptualization: IS, MD, DRG. Methodology: IS, DRG, TB, ES, JD, MD. Investigation: TB, ES, CS, IS. Visualization: TB, ES, IS. Funding acquisition: IS, MD, DRG. Project administration: IS, MD, DRG. Supervision: IS, MD, DRG. Writing – original draft: IS, TB, ES. Writing – review & editing: MD, DRG, JD, IS

## Funding

## Competing interests

The authors declare no competing interests.
