## [Transparent Peer Review file · Communications Biology]

Aberrant choroid plexus formation drives the development of treatment-related brain toxicity

Corresponding Author: Professor Marco Durante

Version 0:

Reviewer comments:

Reviewer #1

(Remarks to the Author)

The study by Bender and Schickel utilized the organoid system to investigate the impact of radiation on brain development. The authors further demonstrated that choroid plexus (CP) formation is affected in animal models upon radiation. To enhance the manuscript's clarity and accuracy, I suggest the following revisions:

- Figure 1C: The section of the organoid appears to be deteriorated, making it difficult to image the structure clearly. The quality of the image needs enhancement. I recommend replacing it with a better image to provide a clear proof of concept for the quality control of organoid generation in the study.
- Writing Clarity: The flow of the writing is sometimes misleading. For example, on page 5, the meaning of the following sentence is unclear: "As d20 organoids still differentiate substantially, changing their cellular composition accordingly, samples were taken at approximately d60. In contrast, the more mature organoids irradiated at d80 were followed up at d100." Please revise this sentence for clarity.
- Missing Data: On page 5, the following sentence lacks data: "In the highest dose cohorts, the size was lower than in the d20 samples (analyzed 40 days prior)." Please provide the missing data.
- Figure Labeling: I suggest more frequent labeling of figure numbers to reference the data described in the results section for better clarity.
- Figure 4: Typically, these organoids differentiate, and a minimal number of progenitors are found. Given that no PAX6/SOX2 is detectable in irradiated organoids, it would be beneficial to quantify the density across organoids to show the neurogenic enrichment compared to the control. If similar density is observed between groups, the rate of gliogenesis should be examined.
- Figure 5: While it is clear that the structure is indeed CP in these organoids, definitive markers of CP such as TTR should be examined at the mRNA level. Additionally, markers for the cortical hem such as MSX1/2 should be examined to determine whether the precursors of CP are being generated or have rapidly differentiated into mature CP. Refer to Figure 1 in this recent paper (DOI: 10.1126/sciadv.adj4735) for reference. If there is an enrichment of cortical hem markers, IHC analysis should also be done for these markers.
- Figure 5C: The authors claim that changes in the color of the cavity indicate fluid production. I am not sure if this is correct. Please provide a reference to support this claim.
- Page 8: The authors claim that "CP formation from neuroepithelial cells and its underlying mechanisms are unclear." This is not accurate, as the mechanisms of CP formation are well-studied. BMP4 and Wnt signaling instruct CP formation at the forebrain level, while Shh instructs CP formation at the hindbrain level (DOI: 10.1126/sciadv.adj4735).
- Radiation Impact: Do the authors suggest that radiation impacts CP formation in rats? If so, is this opposite to what is described in the organoid settings, where more CP is induced upon radiation? Please clarify.

Reviewer #2

(Remarks to the Author)

In the present manuscript Bender et al., studied post-irradiation long-term consequences in hES-derived cerebral organoids. By mainly using a combination of immunofluorescence and qRT-PCR, the authors demonstrated that when immature organoids (d20) are exposed to radiation it will induce the formation of wide cavities filled with a liquid-like substance that is

actively produced by the organoid. The authors data suggest that radiation drives the differentiation of choroid plexus cells, most likely by activation of Notch and Wnt signaling pathways. This study is of great interest and of immediate relevance for the biomedical field. It presents original conclusions that could have a meaningful impact within the scientific community. However, in the present form is not acceptable, it requires major revisions.

Major comments

- I. Introduction is quite unbalanced. It is initially focused on glioblastoma, giving the wrong impression that radiotherapy is only used for these types of tumors. However, it is then expanded to brain tumors, but it became unclear whether the introduced RN-BBD data are only related to glioblastoma. The concept of pediatric brain tumors is only briefly introduced at the end of the introduction. Similarly, the discussion is general, vague, not critically written and in many parts speculative. I would strongly encourage the authors to tailor the discussion and to not speculate on their data.
- II. Figure 1 should contain both d20 and d80 stained organoids. Best would be to have images showing the entire organoid and how they change over time. This is true also for Figure 4, whole organoids should be visible or at least most of them.
- III. What does the author mean with "maximum diversity of initial progenitors" (lane 84). It would be great if the authors could specify to which progenitors they are referring to.
- IV. Was MEA performed at both d20 and d80? If so, please share the data. The activity shown in Supplementary Figure 1 is very low for a d80 cortical organoid. Do you think this is due to low density of induced neurons or immature state of the derived neurons?
- V. Why size measurement is a good read-out to evaluate the postirradiation consequences? Rationale is missing.
- VI. Introduction to figure 2 focuses on a subset of radiation protocol (lines 96-101), however more types of radiation protocols (SOBP and 10/15 Gy) are introduced a few sentences below (lines 107-108). I would strongly encourage the authors to homogenize their intro and clearly present the used protocols including their rationale.
- VII. Figure 2 compares organoids irradiated at d20 (immature) and d80 (mature), however for the immature organoids the authors waited up to 40 days for the phenotypical analysis whereas for the mature organoid authors only waited 20 days. How did the authors select the timeline? Did the authors check whether a longer waiting time for the mature organoid could worsen the postirradiation phenotype? Is Figure 2A left panel missing a sham irradiated control? It would be great to have the average areas also for the control, or the authors could include the % of differences between sham and irradiated organoids.
- VIII. Please specify whether LDH was measured at d60 for the immature organoids and d100 for the mature one.
- IX. Why LDH is increased in sham controls and decreased in irradiated organoids? Based on the rationale of the paper, I would have expected the opposite considering LDH amount was used as a proxy of necrosis. Are the authors suggesting there is less necrosis in immature organoids? This is in contrast with the information given in the introduction section. The authors state $n = 3$, meaning only 3 organoids were measured? Was this experiment repeated multiple times?
- X. Line 119 states "In accordance with the growth retardation..." it is unclear to me to what the authors are referring to
- XI. Line 121 states "...it was more effective.." was the measurement statistically different but the authors forgot to mention? If this is not the case, I would be more careful to describe the results. This figure also contains $n = 2-3$ and $n = 10$, please specify the panels to which this is referring to.
- XII. Title of Supplementary Table 1 is "Quantitative analysis of the dose-response curve for liquid-filling cavities". The authors however forgot to define what they are quantifying, and the data do not match data showed in figure 3.
- XIII. Line 130-131 states "...we examined the distribution of neuroepithelial stem cells in organoids before and after irradiation." I find this reasoning quite weak and untrue for the consequential analysis performed. Authors didn't analyze neuroepithelial lineage in their organoids, so I would strongly encourage them to be more specific and state clear fact.
- XIV. The analysis presented in figure 5 is not enough to conclude that the structure described is choroid plexus. First, it would be necessary to include lineage markers, as *Otx2* and *Lmx1a*, and a wider panel of differentiated and unique choroid plexus markers (for example *TTR*, *CLIC6*, *claudin 1*, *HTR2C*). In addition, I would encourage the authors to include low magnification image showing how much of the organoids is potentially positive for choroid plexus-like markers.
- XV. What is the hypothesis related to IGF2? Why is it upregulated in irradiated organoids and only in mature one?
- XVI. RT-qPCR analysis of Notch is predominantly not statistically different between the two conditions and the potential differences, if any, are minimal. I find it frustrating that the authors conclude with a narrative that is not supported by their data. In addition, I would like to point out that choroid plexus epithelial cells do not express and release Wnt ligands (only exception known is *wnt5B* from 4V in mouse). The wnt ligands are usually produced and released by hems (as cortical hems and rhombic lip) whereas the choroid plexus cells release BMP proteins. Therefore, it is unclear to me how an upregulation of mRNA of wnt ligands should be interpreted as a supportive data for the presence of choroid plexus-like cells. To me, the wnt ligands analysis support a the presence of hem-like cells. So, a detailed analysis of progenitor markers would be critical to solve this conundrum.
- XVII. Low magnification image of potential claudin-3 and aquaporin1 staining in cortical area (Figure 7) in an irradiated rat, it is definitively not enough to claim cell fate change. I do think this experiment is quite crucial for the paper, but if the authors decide to keep these data set in the paper, I would strongly encourage them to expand the analysis, test for lineage markers, quantify the data between independent animals, prove that the staining is specific and more.

Minor comments

- Abstract is a concise summary of the data, and it should not offer interpretation
- I would strongly encourage the authors to not abbreviate the cortical organoid as CO.
- The representative image selected for Figure 1 panel C gives the impression the organoids are either unhealthy or the authors selected a small area for not showing complete distribution of SOX2/Nestin
- Figure 4: it was cut off on the left margin and it is therefore not readable
- I would strongly encourage the authors to include RREDI for all available antibodies and remove the column called datasheet

Version 1:

Reviewer comments:

Reviewer #1

(Remarks to the Author)

I have carefully reviewed the authors' responses and revisions, including the additional figures, improved clarity in writing, and supplementary data addressing several of my original comments. Their efforts have enhanced the manuscript considerably. Specifically, the authors replaced suboptimal images, clarified ambiguous statements, included missing data, and offered more detailed explanations of the rationale and methods employed. These revisions have substantially improved the overall clarity and presentation of the research. While the authors have addressed many of the concerns raised, I believe the study remains somewhat primitive and lacks additional experiments and models (animal models, transcriptomic etc) to support its findings comprehensively.

Reviewer #2

(Remarks to the Author)

The authors constructively addressed most of the raised comments. I would, however, strongly encourage the authors to improve the aesthetically presentation of the figures. In particular, figure 2, 4 and 5 should be compacted and presented all together to improve their readability and the understanding of the data.

I also would like to suggest to change some wording within the abstract, like "...IGF2 mRNA found predominantly in cerebrospinal fluid". mRNA experiments were performed from organoid preparation and not from CSF, so it is not clear to what exactly the authors are referring to. Lastly the authors concluded that "We conclude that radiation-induced brain image changes can be attributed to aberrant CP formation, providing a new cellular mechanism and strategy for possible countermeasures." I do not fully agree with this conclusion because the authors did not demonstrate that the aberrant CP formation are responsible of the radiation-induced brain image changes. I would like to suggest to the authors to be more specific in their statements.

Overall, in the present form, the manuscript should be accepted for publication with minor revision.

Answers to the reviewer

We are deeply grateful to the reviewers for the supportive comments and important suggestions that have helped us to improve our manuscript considerably. With additional experiments and analyses we tried to address these comments and suggestions in the revised version of our manuscript as much as possible. Changes in the text are written in red color to be easily identifiable. We hope that in its revised form the manuscript will now be suitable for publication in *Communications Biology*.

To reviewer 1:

1. Figure 1C: The section of the organoid appears to be deteriorated, making it difficult to image the structure clearly. The quality of the image needs enhancement. I recommend replacing it with a better image to provide a clear proof of concept for the quality control of organoid generation in the study.

We replaced Figure 1B and 1C with Supplementary Figure 1A and 1B. They show the characterization of immature organoids at d20 compared to mature organoids at d80 using immunofluorescence analyses of SOX2, NES, PAX6 and MAP2 in cross-sections of the entire organoids as well as in magnifications of regions of interest.

2. Writing Clarity: The flow of the writing is sometimes misleading. For example, on page 5, the meaning of the following sentence is unclear: "As d20 organoids still differentiate substantially, changing their cellular composition accordingly, samples were taken at approximately d60. In contrast, the more mature organoids irradiated at d80 were followed up at d100." Please revise this sentence for clarity.

We changed this sentence and wrote "Given that d20 organoids still differentiate substantially, changing their cellular composition accordingly, samples were taken approximately 40 days later (at d60) to allow for the observation of radiation impacts on cell differentiation. In contrast, the more mature organoids, where no dramatic changes in cell differentiation and thus cell composition occur, were irradiated at d80 and followed up at d100".

3. Missing Data: On page 5, the following sentence lacks data: "In the highest dose cohorts, the size was lower than in the d20 samples (analyzed 40 days prior)." Please provide the missing data.

The missing data was added, and an outdated version of the plot (Figure 2A) was replaced.

4. Figure Labeling: I suggest more frequent labeling of figure numbers to reference the data described in the results section for better clarity.

We included more labels of the figures in the text where the respective results were described.

5. Figure 4: Typically, these organoids differentiate, and a minimal number of progenitors are found. Given that no PAX6/SOX2 is detectable in irradiated organoids, it would be beneficial to quantify the density across organoids to show the neurogenic enrichment compared to the control. If similar density is observed between groups, the rate of gliogenesis should be examined.

We included low-magnification images in Figure 4 with respective magnifications and additional qPCR data. Our data support the presence of neural stem cells (positive for SOX2 and NES) and neural progenitors (PAX6 positive) that still can be found in some areas of irradiated organoids including proliferation zones at d66 indicating a more immature differentiation status. Also, neurogenic enrichment is evident in more mature organoids from images showing MAP2 expression. In more mature organoids at d100 (irradiated at d80), proliferation zones were rarely observed with no or weak expression of SOX2, slightly higher expression of NES and PAX6, and predominant expression of MAP2. Considering that the glia cells (i.e. astrocytes and oligodendrocytes) are a minor cell population in cerebral organoids generated by unguided Lancaster protocol (doi: 10.1038/nature12517), and maturation of these cells occurs at much later time points (beyond d100), the effect of radiation on gliogenesis was not at the focus of our research. Correspondingly, Supplementary Figure S4 shows very scarce GFAP signal in sham controls and irradiated organoids.

6. Figure 5: While it is clear that the structure is indeed CP in these organoids, definitive markers of CP such as TTR should be examined at the mRNA level. Additionally, markers for the cortical hem such as MSX1/2 should be examined to determine whether the precursors of CP are being generated or have rapidly differentiated into mature CP. Refer to Figure 1 in this recent paper (DOI: 10.1126/sciadv.adj4735) for reference. If there is an enrichment of cortical hem markers, IHC analysis should also be done for these markers.

We refrained from analyzing TTR, even though it is produced in the CP and one of the most frequent proteins in the cerebrospinal fluid, as others found protein depositions reactive to anti-TTR antibodies in the brain parenchyma (<https://doi.org/10.1186/1750-1326-6-79>). This could be explained by neuronal endocytosis of TTR synthesized in the CP. However, this could obscure signals from newly developed CP cells in the cortical tissue. In mRNA analyses, TTR would not enhance our knowledge about the maturity of the newly generated CP. Instead, we analyzed the mRNA expression of inward-rectifying K⁺ channel, KIR7.1, which is highly expressed in CP epithelium (DOI: 10.1186/1743-8454-4-8) and contributes to the formation of CSF. This transporter is enriched in adult CP (doi: 10.1016/j.cell.2021.04.003), allowing to proof the existence of mature CP cells in the cerebral organoids. In addition, we included qPCR analyses of the cortical hem marker MSX1, LMX1, which is expressed by cortical hem and CP, and OTX2, a marker for CP epithelium.

Strong expression of MSX1 in the irradiated organoids indeed points to the formation of cortical hem prior to CP development. Likewise, LMX1A and OTX2 are increased as well, albeit less pronounced. Choroid epithelium formation was also demonstrated by IF analyses of ZO1. Unfortunately, due to the time constraint for supplementary irradiation experiments, we could not provide additional IF analyses for these markers.

7. Figure 5C: The authors claim that changes in the color of the cavity indicate fluid production. I am not sure if this is correct. Please provide a reference to support this claim.

We mention that we observed the exclusion of the ink from the cavity in the ink test (now please refer to Figure 5F). This test, shown in P. Coyle, Spatial Features of the Rat Hippocampal Vascular System, Exp. Neurology 58, 549-561, 1978 or M.-H. Tsai et al., A Mouse Model for the Study of Vascular Permeability Changes Induced by Arsenic, Toxicology Mechanisms and methods 15, 433-437, 2005, serves to prove an intact barrier function. In case of a leaky barrier, the ink would not be excluded from the cavity.

8. Page 8: The authors claim that "CP formation from neuroepithelial cells and its underlying mechanisms are unclear." This is not accurate, as the mechanisms of CP formation are well-studied. BMP4 and Wnt signaling instruct CP formation at the forebrain level, while Shh instructs CP formation at the hindbrain level (DOI: 10.1126/sciadv.adj4735).

We apologize for this statement and removed the incorrect sentence. We agree and are aware that the mechanisms of the CP formation are well described in the literature that we partially had already included in the manuscript elsewhere. We indeed used this literature a basis to analyze changes in the WNT signaling and now also included BMP4.

9. Radiation Impact: Do the authors suggest that radiation impacts CP formation in rats? If so, is this opposite to what is described in the organoid settings, where more CP is induced upon radiation? Please clarify.

From the data obtained in the organoid system, we propose that radiation in rare cases induces focal CP formation from neuroepithelial stem cells leading to the typical frond-like structures with increasing CSF production that have great resemblance to the contrast-enhanced lesions (CEL) seen in MRI scans of patients who received cerebral radiotherapy. This proposition is indeed supported by MRI studies of glioma patients that have been treated with protons showing a higher incidence of CEL and thus lower dose tolerance of the periventricular region compared to others (10.1016/j.radonc.2022.11.011 and 10.1016/j.ijrobp.2020.03.013). We attempted to verify our proposition also by analyzing brain tissue from a previous animal experiment. However, due to the limited tissue availability and inability to perform sufficient experimental and statistical analyses, we removed those data and plan a follow-up study in rodents to specifically address this question.

To reviewer 2:

1. Introduction is quite unbalanced. It is initially focused on glioblastoma, giving the wrong impression that radiotherapy it is only used for these types of tumors. However, it is then expanded to brain tumors, but it became unclear whether the introduced RN-BBD data are only related to glioblastoma. The concept of pediatric brain tumors is only briefly introduced at the end of the introduction. Similarly, the discussion is general, vague, not critically written and in many parts speculative. I would strongly encourage the authors to tailor the discussion and to not speculate on their data.

We adjusted the introduction focusing more on the actual issue addressed in our study, namely elucidating the mechanisms leading to contrast enhanced lesions in the brain of irradiated patients. This phenomena, also referred to as radiation necrosis (RN) is not restricted to a certain tumor type, but rather the irradiation scheme. Pediatric patients are of special concern in this respect as they develop unexpectedly high rates of RN. We added further analyses to substantiate our initial hypothesis and updated the discussion with the new findings and provided a critical analysis of our data with additional references.

2. Figure 1 should contain both d20 and d80 stained organoids. Best would be to have images showing the entire organoid and how they change over time. This is true also for Figure 4, whole organoids should be visible or at least most of them.

We replaced Figure 1B and 1C with Supplementary Figure S1A and S1B. They show the characterization of immature organoids at d20 compared to mature organoids at d80 using immunofluorescence analyses of SOX2, NES, PAX6 and MAP2. In Supplementary Figure S1A and S1B as well as in Figure 4 cross-sections of the entire organoids as well as magnifications of regions of interest are shown in better quality.

3. What does the author mean with “maximum diversity of initial progenitors” (lane 84). It would be great if the authors could specify to which progenitors they are referring to.

The rationale for using an unguided differentiation protocol was the fact, that CELs occur randomly with no preference of certain brain regions. Thus, we attempted to interfere as little as possible with the formation of the various cells types generated from neural stem cells. We changed the wording in the text to “to generate cerebral organoids in which neural progenitors are allowed to differentiate spontaneously with a maximum diversity.”.

4. The activity [MEA measurement] shown in Supplementary Figure 1 is very low for a d80 cortical organoid. Do you think this is due to low density of induced neurons or immature state of the derived neurons?

Calcium imaging and IF staining of neurofilaments using an SMI312 antibody reveal a rich neuronal network in the organoids used for this study (data not shown). However, MEA measurements with possibly suboptimal chips always showed low activity most likely misrepresenting the actual maturity. We were able to slightly improve this by using organoid slices, but we also are in the process of applying state-of-the-art MEA meshes, that are currently developed specifically for the use with whole organoids and will be released in early 2025. In expectation of this improved MEA system, we are refraining from publishing the MEA data and removed them from the supplements.

5. Why size measurement is a good read-out to evaluate the postirradiation consequences? Rationale is missing.

In this study, we used d20 and d80 cerebral organoids representing, at most, the embryonic/fetal and early postnatal brain, respectively. Prenatal radiation exposure leads to growth retardation and small head/brain size (reviewed in B. Yang et al., <https://doi.org/10.1016/j.braindev.2016.07.008>). Furthermore, a decrease in the brain volume due to the loss of white matter after radiotherapy was reported in children and young adults with low-grade astrocytoma and medulloblastoma, and it could be associated with a decrease in cognitive functions, e.g. a lower IQ ([doi.org/10.1016/S0730-725X\(98\)00014-9](https://doi.org/10.1016/S0730-725X(98)00014-9); doi: 10.1002/1531-8249(199912)46:6<834::aid-ana5>3.0.co;2-m). Thus, we hypothesized the d20 organoids, representing a prenatal brain, to be very sensitive to irradiation, more

than mature organoids, responding with growth retardation reflected in a smaller organoid size. We added this rationale to the results part (lines 113-114).

6. Introduction to figure 2 focuses on a subset of radiation protocol (lines 96-101), however more types of radiation protocols (SOBP and 10/15 Gy) are introduced a few sentences below (lines 107-108). I would strongly encourage the authors to homogenize their intro and clearly present the used protocols including their rationale.

We added information on which protocols were used for immature and which for mature organoids including the rationale for their use. We used more doses in each protocol (X-rays and proton irradiation) in the case of the size/ circular area measurement (Fig. 2A and B) and for the quantification of the organoid cavities (Fig. 3 B-C). We also added qualities and doses of irradiation to the scheme in Fig.1.

7. Figure 2 compares organoids irradiated at d20 (immature) and d80 (mature), however for the immature organoids the authors waited up to 40 days for the phenotypical analysis whereas for the mature organoid authors only waited 20 days. How did the authors select the timeline? Did the authors check whether a longer waiting time for the mature organoid could worsen the postirradiation phenotype? Is Figure 2A left panel missing a sham irradiated control? It would be great to have the average areas also for the control, or the authors could include the % of differences between sham and irradiated organoids.

Immature organoids (irradiated at d20) still undergo differentiation to a higher extent than mature organoids (irradiated at d80) and for this reason, we decided to follow immature organoids longer (40 days) to allow for the observation of radiation impacts. For mature organoids, a shorter time (20 days after irradiation) was chosen as no further significant changes took place after day 80 of organoid culture. We included this explanation for the selected timeline in the introduction part of Results (lines 91-96). Mature organoids were followed up to day 140 (60 days after irradiation) for the measurement of the extracellular LDH level as a measure for necrosis. We neither observed significant changes in the LDH level between organoids at d100 and d140 (Figure 2F) nor the additional formation of cavities beyond d100.

Figure 2A was an outdated version. We included the sham-irradiated control at d20 as a dotted line and the % of differences between sham and irradiated organoids in the text.

8. Please specify whether LDH was measured at d60 for the immature organoids and d100 for the mature one.

LDH was measured on d21, d40 and d60 for immature organoids, 1d, 20d and 40d after irradiation, and on d81, d100 and d140 for mature organoids, 1d, 20d and 60d after irradiation, respectively. This information was included in Figure 2E and F and we added more information in the text (lines 138-141).

9. Why LDH is increased in sham controls and decreased in irradiated organoids? Based on the rationale of the paper, I would have expected the opposite considering LDH amount was used as a proxy of necrosis. Are the authors suggesting there is less necrosis in immature organoids? This is in contrast with the information given in the introduction section. The authors state n = 3, meaning only 3 organoids were measured? Was this experiment repeated multiple times?

We agree with the reviewer that the expected result is an increase in the extracellular LDH, a marker of necrosis, as an effect of irradiation. However, our results do not follow the general hypothesis that irradiation leads to high levels of necrosis. The observed increase in LDH level predominantly in sham controls was rather correlated with an increase in the organoid size during the culture time when a necrotic core is formed due to the insufficient supply of nutrients and oxygen in larger organoids, and this was already described in the literature (Lancaster et al., 2017; 10.1038/nbt.3906). This is in accordance with the reviewer's point that immature and smaller organoids have less necrosis than mature and larger organoids. In the introduction, we write about RN (radiation necrosis) which is a term sometimes mistaken for contrast-enhanced lesions observed post-irradiation. Additional staining against Ki-67 revealed that irradiation led to a decrease in cell proliferation in young organoids, and little or no change in mature organoids that in general exhibited a lower number of proliferating cells. A dose-dependent increase in apoptosis, most prominent 1d after irradiation, was evident by the expression of the activated caspase-3 in both young and mature organoids, and it was overall more pronounced in young organoids.

The LDH assays were performed using two to three independent experiments with 10 organoids per variant/group. We included the meaning of N (number of independent experiments) and n (number of organoids per variant/group) in the figure legends and methods.

10. Line 119 states “In accordance with the growth retardation...” it is unclear to me to what the authors are referring to.

We added more information and a reference to Figure 2A for better clarity as we intended to make a connection between the growth retardation of organoids and the formation of liquid-filled cavities in response to irradiation. While we observed a significant dose-dependent growth retardation after irradiation, there was an increase in the % of organoids with cavities also in the dose-dependent manner when using the same irradiation protocols.

11. Line 121 states “...it was more effective..” was the measurement statistical different but the authors forgot to mention? If this is not the case, I would be more careful to describe the results. This figure also contains n =2-3 and n=10, please specify the panels to which this is referring to.

For better clarity, we included new plots (non-fitted data) showing significant changes in Figure 3 (statistical tests used for this assay are stated in the figure legend), and a plot with fitted data, used to calculate the relative biological effectiveness, RBE, (Supplementary Table 1), is now Supplementary Figure S3.

12. Title of Supplementary Table 1 is “Quantitative analysis of the dose-response curve for liquid-filling cavities”. The authors however forgot to define what they are quantifying, and the data do not match data showed in figure 3.

We changed the wording in the table title to “Quantitative analysis of the dose-response curve for the percentage of organoids with liquid-filling cavities”. Data in Supplementary Table 1 contain values of the linear fit of the data from Figure 3B-C. These values were used to calculate the relative biological effectiveness (RBE) of X-rays and protons, respectively, for the cavity formation (percentage of organoids containing liquid-filled cavities). These data show that protons in the SOBP are more effective in the formation of cavities than X-rays or protons in the plateau. We also included new plots (non-fitted data) in Figure 3 and moved a plot with fitted data to the Supplements (Figure S3).

13. Line 130-131 states “...we examined the distribution of neuroepithelial stem cells in organoids before and after irradiation.” I find this reasoning quite weak and untrue for the consequential analysis performed. Authors didn’t analyze neuroepithelial lineage in their organoids, so I would strongly encourage them to be more specific and state clear fact.

We agree with the reviewer, and we corrected this sentence by mentioning specific markers of cells we analyzed including markers of precursors of CP (hem) and progenitors of neurons.

14. The analysis presented in figure 5 is not enough to conclude that the structure described is choroid plexus. First, it would be necessary to include lineage markers, as Otx2 and Lmx1a, and a wider panel of differentiated and unique choroid plexus markers (for example TTR, CLIC6, claudin 1, HTR2C). In addition, I would encourage the authors to include low magnification image showing how much of the organoids is potentially positive for choroid plexus-like markers.

We performed qPCR analyses of additional markers such as MSX1, OTX2, and LMX1A, and KIR7.1. For the rationale using the latter rather than TTR, please refer to the comments to reviewer 1, item 6. We included low-magnification images in Figure 5. The differentiated epithelial sheet also expresses AQP1 at the apical side and CLDN3 at the lumen, and we could show their expression at both the mRNA and the protein level. Although AQP1 is also expressed in endothelial cells, and CLDN3 is expressed in the vessels of developing rat brain, in cerebral organoids generated by Lancaster protocol (doi: 10.1038/nature12517), no endothelial cells and blood vessels are formed and their expression is restricted to the epithelia of the choroid plexus. Choroid epithelium formation was also demonstrated by IF analyses of ZO1. Unfortunately, due to the time constraint for supplementary irradiation experiments, we could not provide additional IF analyses for all markers, analyzed in the qPCR assays.

15. What is the hypothesis related to IGF2? Why is it upregulated in irradiated organoids and only in mature one?

Embryonic as well as adult CSF contains IGF2, a factor that supports the propagation of neural stem cells and thus peaks during brain development. In our study, it is increased after irradiation irrespective of the age of the irradiated organoids (Figure 5A). An excess CSF production as seen in this study therefore can pave the way for neural stem cell propagation and subsequent differentiation possibly leading to the regeneration of the radiation-damaged neuronal network.

16. RT-qPCR analysis of Notch is predominantly not statistically different between the two conditions and the potential differences, if any, are minimal. I find it frustrating that the authors conclude with a narrative that is not supported by their data. In addition, I would like to point out that choroid plexus epithelial cells do not express and release Wnt ligands (only exception known is wnt5B from 4V in mouse). The wnt ligands are usually produced and released by hems (as cortical hems and rhombic lip) whereas the choroid plexus cells release BMP proteins. Therefore, it is unclear to me how an upregulation of mRNA of wnt ligands should be interpreted as a supportive data for the presence of choroid plexus-like cells. To me, the wnt ligands analysis support a the presence of hem-like cells. So, a detailed analysis of progenitor markers would be critical to solve this conundrum.

We corrected these sentences in the text and described our results carefully. The expression of WNT ligands at the mRNA level suggests that hem-like cells are present and secrete these ligands. Since the formation of cortical hem precedes the formation of CP (DOI: 10.1126/sciadv.adj4735; reviewed in: <https://link.springer.com/article/10.1007/s00018-022-04314-1>), we also analyzed the expression of lineage markers, LMX1A and OTX2, and more mature markers of CP such as AQP1, CLDN3, IGF2. We also included the analysis of BMP4 which is necessary and sufficient for the specification of choroid epithelia (<https://doi.org/10.1523/JNEUROSCI.3227-12.2012>).

17. Low magnification image of potential claudin-3 and aquaporin1 staining in cortical area (Figure 7) in an irradiated rat, it is definitively not enough to claim cell fate change. I do think this experiment is quite crucial for the paper, but if the authors decide to keep these data set in the paper, I would strongly encourage them to expand the analysis, test for lineage markers, quantify the data between independent animals, prove that the staining is specific and more.

From the data obtained in the organoid system, we propose that radiation in rare cases induces focal CP formation from neuroepithelial stem cells leading to the typical frond-like structures with increasing CSF production that have great resemblance to the contrast-enhanced lesions (CEL) seen in MRI scans of patients who received cerebral radiotherapy. This proposition is indeed supported by MRI studies of glioma patients that have been treated with protons showing a higher incidence of CEL and thus lower dose tolerance of the periventricular region compared to others (DOI:10.1016/j.radonc.2022.11.011 and DOI: 10.1016/j.ijrobp.2020.03.013). We attempted to verify our proposition also by analyzing brain tissue from a previous animal experiment. However, due to the limited tissue availability and inability to perform sufficient experimental and statistical analyses, we removed those data and, in agreement with the reviewers' advice, plan a follow-up study in rodents to expand the analysis, test for lineage markers, quantify the data between independent animals and prove that the staining is specific.

Minor comments

18. Abstract is a concise summary of the data, and it should not offer interpretation.

We revised the abstract according to the reviewers' suggestions included more background and context of work, major results only and conclusion.

19. I would strongly encourage the authors to not abbreviate the cortical organoid as CO.

We now changed it and used "cerebral organoid" instead of CO.

20. The representative image selected for Figure 1 panel C gives the impression the organoids are either unhealthy or the authors selected a small area for not showing complete distribution of SOX2/Nestin.

We replaced Figures 1B-C with Supplementary Figure S1. Here we show the complete distribution of SOX2, NES, PAX6 and MAP2 in sections of the entire organoids and magnifications of regions of interest in higher quality pictures.

21. *Figure 4: it was cut off on the left margin and it is therefore not readable.*

Figure 4 was an outdated version, and it was replaced with a new appropriate version again showing the complete distribution of SOX2, NES, PAX6 and MAP2 in sections of the entire organoids and magnifications of regions of interest in higher quality pictures in Figure 4B-C and E-F, respectively, in addition to qPCR analyses (Figure 4A and D) corroborating the IF analyses.

22. *I would strongly encourage the authors to include RRDI for all available antibodies and remove the column called datasheet.*

We deleted the datasheet column and included RRDI column for all antibodies.

To reviewer 3

1. *The manuscript presents an insightful study of the effects of irradiation on cerebral organoids. However, to strengthen the findings and enhance the robustness of the conclusions, we recommend the inclusion of additional controls that consider sex-specific differences. Specifically, I suggest incorporating male iPS cells together with female ES cell lines to generate cerebral organoids. This addition is crucial due to the documented differences in survival rates and responses to treatment between males and females in glioblastoma (GBM) patients (Dmukauskas M et al., Journal of Neuro-Oncology, 2024).*

The reviewer raises an important point. Indeed, as reviewed by A. Carrano et al. (doi.org/10.3390/cells10071783, 2021), the incidence of GBM is 1.6 times higher in men than in women and there are gender-specific differences in the genetic and molecular mechanisms associated with GBM. Likewise, women have a longer survival and better outcomes, e.g. M. Tian et al. (DOI: 10.1042/BSR20180752, 2018) observed a difference in the 5-year cancer-specific survival (CSS) rates in the male and female groups, which were 6.8% and 8.3%, respectively (P=0.002 by univariate and P<0.001 by multivariate analysis). There are also sex specific differences in adverse events occurring after a GBM diagnosis (Dmukauskas M et al., Journal of Neuro-Oncology, 2024). However, adverse events (AE) regarding the Nervous System, albeit being the most common AE, were not sex-specific. This is also reflected in radiation necrosis-related studies: E.g. J. Kerschbaumer et al. (doi.org/10.3390/cancers13194736) found a higher tumor diameter and higher radiation dose associated with an increased risk of radiation necrosis, but no gender influence. Similarly, E.J. Lehrer et al. did not find a significant gender difference to develop treatment-related imaging changes in an international multicenter study of 697 patients (DOI: 10.3171/2022.7.JNS22752, 2023). Eulitz et al. observed an increased radiosensitivity within the periventricular region as well as a spatial correlation of radiation induced brain injury with an increased relative biological effectiveness (RBE), but again no gender influence. Therefore, as age, radiation quality and dose seem to be more important influencers of RN and the generation of cerebral organoids is feasible gender-independently (E. K. Stachowiak et al., <https://doi.org/10.1038/s41398-017-0054-x>), we refrained from using male stem cell lines in addition to the female ones. Additional data, that have become available and show findings similar to ours in mice, have been cited as well (J. H. Ribeiro et al., bioRxiv preprint doi: <https://doi.org/10.1101/2024.06.27.600564>).

2. *Studying the effects of treatments like irradiation on cerebral organoids that mimic healthy brain tissue does offer insights into potential collateral damage to normal brain cells, which is valuable. However, it does not fully replicate the complexities of the tumor microenvironment found in glioblastoma, which can significantly influence treatment outcomes and side effects.*

We fully agree with this statement and have broadened our organoid models to also include a brain tumor model, induced via the overexpression of an oncogene. This enabled us to observe changes at the border between tumor-like and normal brain organoid tissue upon irradiation. The respective study will be submitted in due time. Here however, we focused on a radiation-induced side effect, that is irrespective of the tumor type and affecting the normal brain tissue. Thus we employed cerebral organoids as the currently best *in vitro* model to mimic inaccessible human brain tissue.

3. “All sham controls showed an increase in size. Organoids irradiated at d20 displayed a significant dose-dependent decrease in size at d60 when compared to the sham-controls. In the highest dose-cohorts, the size was lower than in the d20 samples (analyzed 40 days prior). The response of mature organoids irradiated on d80 was less drastic than seen for the d20 organoids. It is recommended to correlate these results with staining for proliferation markers, such as Ki67, or apoptosis markers, such as Caspase-3 and TUNEL, to gain insights into the mechanisms governing irradiation-mediated resizing of the organoids.”

We performed additional staining against Ki-67 (shown in Figure 2C and D) and could correlate a decrease in the organoid size with a decrease in cell proliferation as evidenced by a dose-dependent decrease in the expression of Ki-67 in young organoids or little or no change in mature organoids with a generally lower number of proliferating cells. Also, we observed a dose-dependent increase in the expression of activated caspase 3 in both young and mature organoids but particularly evident in young organoids (shown in Supplementary Figure S2).

4. “Measuring necrosis via lactate dehydrogenase (LDH) release in samples irradiated with X-rays at d20 revealed highest rates of necrosis in the sham controls, but significantly lower levels in the irradiated samples 20 and 40d after exposure. In samples irradiated at d80, no significant changes could be observed.” As organoids grow larger, cells in the inner core may become deprived of nutrients and oxygen, leading to cell death and necrosis. Since organoids lack a vascular system to efficiently supply cells with essential nutrients and oxygen, the formation of necrotic regions is exacerbated as organoid size increases. If irradiation causes organoids to become smaller, they may be less prone to developing necrotic cores compared to untreated, larger organoids. Smaller irradiated organoids with less necrosis might release less LDH into the medium, as LDH release is a marker of cell damage and death. Therefore, a reduction in LDH release in irradiated organoids could be mistakenly interpreted as a protective or beneficial effect of irradiation when it might simply reflect the smaller size and reduced necrosis. To accurately assess the effects of irradiation, it is crucial to account for potential differences in necrotic core formation between irradiated and control organoids. One approach is to use metrics that normalize LDH release to the viable cell number or organoid size. Including additional markers of cell viability and necrosis, alongside LDH, will provide a more accurate understanding of the effects of irradiation.

We indeed agree with the reviewer that the reduced LDH release in irradiated organoids is not due to a possible protective/beneficial effect of the radiation, but rather due to the radiation-induced reduction in size. In line with the reviewers’ argumentation, we included IF analyses of the proliferation marker Ki-67 and of active Caspase-3 to analyze apoptosis (please refer to our answer to Reviewer 2, item 9).

5. “The ink was completely excluded from the cavities, proving intact barrier characteristics over the observed time of up to 5 min (Figure 5C)”.

Without quantifiable data, it is difficult to state about the integrity of the barrier. Visual inspection may not be sensitive enough to detect small but significant changes in permeability. We recommend other assays commonly used to test barrier permeability, such as dextrans.

We agree with the reviewer that the use of dextrans of varying molecular weights as reported in Pellegrini et al. (Science 369, 159-171, 2020) would allow quantifying the permeability of the barrier. However, the ink test is perhaps the oldest and easiest way to qualitatively proof barrier function as e.g. shown in P. Coyle, Spatial Features of the Rat Hippocampal Vascular System, Exp. Neurology 58, 549-561, 1978 or M.-H. Tsai et al., A Mouse Model for the Study of Vascular Permeability Changes Induced by Arsenic, Toxicology Mechanisms and methods 15, 433-437, 2005. As we did not intend to measure

the permeability of the CP-like barrier in correlation to various molecular weights, but to merely show a general barrier function, we chose the ink assay as an easy approach that can be even judged by the naked eye. At a mass of 0.76 kDa the blue ink used in our study should have penetrated and pointed towards any breach within the CP-like epithelial lining as easily as the 3-70 kDa dextrans, used in the Pellegrini study.

5. The text contains some theoretical inaccuracies: 1) “The choroid plexus (CP) is comprised of highly fenestrated epithelial tissue that forms the blood-cerebrospinal fluid (CSF) barrier and is the production site for CSF in the vertebrate brain”. The error in the sentence lies in the description of the choroid plexus (CP) as “highly fenestrated epithelial tissue.” The correct anatomical detail is that the choroid plexus comprises epithelial cells covering a core of highly vascularized connective tissue (not fenestrated epithelial tissue). The choroid plexus capillaries are fenestrated, allowing for the exchange of substances necessary to produce cerebrospinal fluid (CSF); 2) “These structures were also positive for the gap junction protein ZO1”, ZO1 is a tight junction protein, not a gap junction protein.

We corrected the inaccurate text and included the reference.

6. The manuscript in its current form does not seem to fulfill the aesthetic requirements for publication. Specifically, Figure 4 is cropped. Additionally, some staining appears unconvincing, particularly the staining with PAX6, a transcription factor, which does not seem to exhibit nuclear staining.

Figure 4 was an outdated version, and it was replaced with a new appropriate version.

7. The authors need to clarify the terminology for reporting the number of experiments (e.g., batches, N) and the number of organoids (e.g., n). Additionally, they should increase the number of organoids and batches to strengthen the findings. Overall, the choice of statistical tests is appropriate for the analysis.

We included the meaning of N (number of independent experiments) and n (number of organoids per variant/group) in figure legends and in the methods.

8. The manuscript demonstrates good levels of quality and consistency in using acronyms uniformly to refer to specific terms. However, attention to grammar, redundancy (e.g., “we we probed,” line 7, page 7), and nomenclature consistency is necessary. Ensure that acronyms maintain correct capitalization throughout the text, such as using “COs” consistently instead of alternating with “Cos” (line 6, page 11). By refining these aspects, the study can provide more reliable insights into the side effects of radiation therapy and contribute more effectively to the development of strategies to mitigate these effects, ultimately supporting the long-term well-being of GBM patients.

We implemented changes in the text as suggested.

The study by Bender and Schickel utilized the organoid system to investigate the impact of radiation on brain development. The authors further demonstrated that choroid plexus (CP) formation is affected in animal models upon radiation. To enhance the manuscript's clarity and accuracy, I suggest the following revisions:

- **Figure 1C:** The section of the organoid appears to be deteriorated, making it difficult to image the structure clearly. The quality of the image needs enhancement. I recommend replacing it with a better image to provide a clear proof of concept for the quality control of organoid generation in the study.
- **Writing Clarity:** The flow of the writing is sometimes misleading. For example, on page 5, the meaning of the following sentence is unclear: "*As d20 organoids still differentiate substantially, changing their cellular composition accordingly, samples were taken at approximately d60. In contrast, the more mature organoids irradiated at d80 were followed up at d100.*" Please revise this sentence for clarity.
- **Missing Data:** On page 5, the following sentence lacks data: "*In the highest dose cohorts, the size was lower than in the d20 samples (analyzed 40 days prior).*" Please provide the missing data.
- **Figure Labeling:** I suggest more frequent labeling of figure numbers to reference the data described in the results section for better clarity.
- **Figure 4:** Typically, these organoids differentiate, and a minimal number of progenitors are found. Given that no PAX6/SOX2 is detectable in irradiated organoids, it would be beneficial to quantify the density across organoids to show the neurogenic enrichment compared to the control. If similar density is observed between groups, the rate of gliogenesis should be examined.
- **Figure 5:** While it is clear that the structure is indeed CP in these organoids, definitive markers of CP such as TTR should be examined at the mRNA level. Additionally, markers for the cortical hem such as MSX1/2 should be examined to determine whether the precursors of CP are being generated or have rapidly differentiated into mature CP. Refer to Figure 1 in this recent paper (DOI: 10.1126/sciadv.adj4735) for reference. If there is an enrichment of cortical hem markers, IHC analysis should also be done for these markers.
- **Figure 5C:** The authors claim that changes in the color of the cavity indicate fluid production. I am not sure if this is correct. Please provide a reference to support this claim.
- **Page 8:** The authors claim that "*CP formation from neuroepithelial cells and its underlying mechanisms are unclear.*" This is not accurate, as the mechanisms of CP formation are well-studied. BMP4 and Wnt signaling instruct CP formation at the forebrain level, while Shh instructs CP formation at the hindbrain level (DOI: 10.1126/sciadv.adj4735).
- **Radiation Impact:** Do the authors suggest that radiation impacts CP formation in rats? If so, is this opposite to what is described in the organoid settings, where more CP is induced upon radiation? Please clarify.